# Recent Advances in Aggregation-Induced Emission Active Materials for Sensing of Biologically Important Molecules and Drug Delivery System

**DOI:** 10.3390/molecules27010150

**Published:** 2021-12-27

**Authors:** Geeta A. Zalmi, Ratan W. Jadhav, Harshad A. Mirgane, Sheshanath V. Bhosale

**Affiliations:** School of Chemical Sciences, Goa University, Taleigao Plateau 403206, India; geetazalmi446@gmail.com (G.A.Z.); ratanjadhav0725@gmail.com (R.W.J.); harshadmirgane@gmail.com (H.A.M.)

**Keywords:** aggregation induced emission (AIE), chemosensors, anion sensing, biological cell imaging, drug delivery system

## Abstract

The emergence and development of aggregation induced emission (AIE) have attracted worldwide attention due to its unique photophysical phenomenon and for removing the obstacle of aggregation-caused quenching (ACQ) which is the most detrimental process thereby making AIE an important and promising aspect in various fields of fluorescent material, sensing, bioimaging, optoelectronics, drug delivery system, and theranostics. In this review, we have discussed insights and explored recent advances that are being made in AIE active materials and their application in sensing, biological cell imaging, and drug delivery systems, and, furthermore, we explored AIE active fluorescent material as a building block in supramolecular chemistry. Herein, we focus on various AIE active molecules such as tetraphenylethylene, AIE-active polymer, quantum dots, AIE active metal-organic framework and triphenylamine, not only in terms of their synthetic routes but also we outline their applications. Finally, we summarize our view of the construction and application of AIE-active molecules, which thus inspiring young researchers to explore new ideas, innovations, and develop the field of supramolecular chemistry in years to come.

## 1. Introduction

The recent development in the field of supramolecular chemistry, especially in chemo-/biosensing, biological cell imaging, and drug delivery systems, has gained a lot of attention due to its high quantum yield and good photostability [1,2,3]. Hence, numerous fluorescent materials have been developed and have attracted the attention of researchers towards the use of the florescent material such as tetraphenylethyne (TPE) derivatives Schiff’s bases, naphthalenediimide (NDI), pyrene, conjugated polymer, and various metal-organic framework (MOFs) and carbon dots (CDs) as a good platform for sensing contaminants and its application in cell imaging and drug delivery systems [4,5,6,7,8].

Luminescence is a spontaneous light emission process that comes from excited electronic states upon absorption of UV-vis light which has received great attention in various fields in chemistry, physics, material science, medicine and biology [9]. Most of the organic luminophores show this property and have been widely used in sensing applications. However, many organic luminophores are studied in dilute solutions, thus, exhibiting very different photophysical phenomena compared with concentrated solutions. This is a common phenomenon where the luminescence is either weakened or quenched at high concentration this effect is known as “concentration quenching” which is caused due to formation of aggregate which is considered as the detrimental process known as aggregation caused quenching (ACQ) [10,11]. The most common conventional luminophore presenting the ACQ phenomenon is *N,N*-dicyclohexyl-1,7-dibromo-3,4,9,10-perylenetetracarboxylic diimide (DDPD) as shown in Figure 1,which emits strong fluorescence in dilute solutions and suffers from ACQ at high concentration or in the aggregated state. An organic luminophore that shows ACQ is considered detrimental and shows very low sensitivity and hence its use is limited [12,13,14]. However, another phenomenon was discovered by Tang’s group in 2001 group showing overcoming of the ACQ process by the so called aggregation induced emission (AIE) phenomenon [15,16,17] which is exactly opposite to that of ACQ, wherein a molecule is initially non-emissive but becomes highly emissive in an aggregated state, as shown in Figure 2**.** There are other non-radiative decay processes such as restricted intramolecular vibration and rotational relaxation responsible in the AIE process. Many research groups have reported AIE-active molecules that can be used for chemical sensors [18,19,20]. Among them, hexaphenylsilole (HPS) is the common example of the AIE active molecule which exhibits enhancement in fluorescence in an aggregate state. The motions involved, such as restriction of intramolecular motion along with rotation and vibration mechanisms in the AIE active phenomenon, are well explained and accepted [21,22]. The AIE luminogens have high photostability, large stoke shift, a photobleaching resistance property, and show high sensing reproducibility [23]. This characteristic makes luminogens a promising candidate for sensing application. Nevertheless, the AIE active luminogens find wide application in various fields acting as an excellent platform for sensing of food contaminants such as toxic cations and anions, veterinary drugs, pesticides, fertilizers, pathogens such as Gram-positive and Gram-negative bacteria, food additives and so on. In addition, the AIE-active luminogens have various applications such as mechanofluorochromism, optical light-emitting device (OLED) application, solar cells, cell imaging, biosensing, and drug delivery applications. The abnormal AIE phenomenon involves different mechanisms. Interestingly, due to the unique photophysical phenomenon of AIE activity, the AIE active luminogens shows different “turn-on” sensing mechanism via various interactions involving electrostatic interactions, hydrogen bonding, van der Waals interactions, and metal-ligand interactions. Another approach is through recognition of analyte with “turn-on” fluorescence via restriction of intramolecular rotation. Other different electron transfer processes involved in sensing mechanism are photoinduced electron (PET), intramolecular charge transfer (ICT) and Forster resonance electron transfer (FRET). The potential proposed mechanism responsible for sensing occurs through different pathways such as the restriction of intramolecular rotation (RIR), hydrogen bond interaction, J-aggregation, molecular planarization, and twisted intramolecular charge transfer (TICT) [24,25,26,27,28,29]. Restricted intramolecular rotation (RIR) and restricted intramolecular vibration (RIV) were merged as restricted intramolecular motion (RIM) regarded as the important mechanism involved for the AIE effect, Figure 3**.**

Although a few review articles were previously published on AIE-active molecules which describe their application in different fields separately, each review has only explored one type of application. Therefore, in this review, we decided to combine the AIE-active luminogens in biological oriented application such as such as sensing of different essential and non-/toxic ions, biological cell imaging, fluorescent biomarker markers and have further extended this to drug delivery systems for controlled release of a drug, which may attract more researchers in the field. 

## 2. Aggregation Induced Emission (AIE) Active Molecules for Sensing Application 

Metal ions play a very important role in various biological and physiological processes occurring in the body and are required by all life forms. Most of the metal ions participate in many biochemical processes, such as material transportation, energy conversion, information transmission, and metabolic regulation [30]. The presence of excess metal ions may lead to serious health issues and may hamper the various metabolic processes. Therefore, there is a need for monitoring, detection, and study of their distribution in individual systems including tissues, cells, as well as organisms. Several methods have been reported for metal ion detection, but few of these methods suffer from drawbacks such as tedious synthesis, cost and non-selectivity. Nevertheless, small organic molecules with high fluorescence properties found to be more advantageous over other reported chromophores due to high sensitivity, selectivity, high quantum yield, simple synthetic routes, easy operation, and real-time detection [31]. There are several essential (Na^+^, K^+^, Ca^2+^, Zn^2+^ and Mg^2+^, Fe^2+^) and non-essential/toxic metals (Hg^2+^, Pd^2+^, Pb^2+^, Cu^2+^, As^3+^ Cr^3+^) and anions (F^−^, Cl^−^, CN^−^, Br^−^). Detection with a lower detection limit with quantification is very important for both the essential and non-essential metal ions due to their presencefrom a biological and environmental point of view. Among the wide range of applications of fluorescence techniques using fluorescent molecules of various cations and anions, is the most important and found to be a promising active platform in various areas.

Among cations, potassium ion is an essential component of the human body that plays a very important role in monitoring and regulating the various physiological functions, such as heart beat, muscle strength, regulates nervous and renal functions [30,32]. The normal concentration of K^+^ in plasma ranges from 3.5 to 5.5 mM; moreover, if the concentration increases to 7.0 mM then it may result in a complication [33]. To quantify the amount of K^+^ ion, Liu et al. synthesized molecular rotors for detection of K^+^ ion [34] bearing G-quadruplex derived from guanine (G)-rich DNA sequences is used as a structural motif. Initially, the crown ethers and cryptands were frequently used as host moieties for sensing K^+^ ions. There are very few AIE active fluorescent probes that have been reported until now. In this regard, Wang and co-workers utilized an AIE strategy and constructed a highly sensitive and selective fluorescent OFF-ON probe via host-guest molecular recognition by functionalizing a novel crown ether on TPE derivative by the thiol via click reaction, yielding TPE-(SH)_4_ and maleimide-functionalized benzo-15-crown-5(B15C5). The probe TPE-(B15C5)_4_, consisted of TPE as the core moiety with four substituted B15C5 units. Its optical properties were studied via the AIE mechanism and found to be highly selective and sensitive towards K^+^ ions [35]. In addition, Lu and coworkers designed TPE modified with a DNA oligonucleotide-based fluorescent probe exhibiting excellent AIE active behavior and a probe showing excellent sensing behavior towards the K^+^ ion. Furthermore, the probe was used for biological cell imaging [36]. Moreover, another novel potassium sensing oligonucleotide derivative via fluorescence resonance electron transfer (FRET) was studied and used for detecting K^+^ in water [37]. The structure of the most commonly synthesized AIE active fluorescent probe is illustrated in Figure 4.

Calcium is another essential metal ion for various physiological functions such as muscle contraction, blood pressure, heart beats, vascular and nerve functioning at the same time calcium major component of the bone structure in the body [38]. The intracellular calcium is stored in mitochondria and the endoplasmic reticulum while most of the calcium is found in bones and teeth. The change in concentration in Ca^2+^ may result in various diseases resulting in obesity and Alzheimer’s disease [39]. The increase in level of Ca^2+^ in the body may lead to several diseases’ cardiac arrhythmias, sarcoidosis, and tuberculosis. Therefore, sensing and monitoring extracellular and intracellular Ca^2+^ is important for diagnosis. In this regard, Gao and co-workers designed an AIE active probe for in situ detection of calcium. The probe SA-4CO_2_Na can distinguish efficiently between the normal and hypercalcemic calcium in the cell. The probe was synthesized by reacting 5-(chloromethyl)-2-hydroxybenzaldehyde with diethyl imino diacetate in the presence of hydrazine monohydrate to give SA-4CO_2_Et which was further reacted in presence of sodium methoxide affords to form SA-4CO_2_Na. The exact mechanism involved in the detection of Ca^2+^ is by the formation of highly emissive fibrillar aggregate via electrostatic and chelating interaction between the iminodiacetate groups and Ca^2+^ ion. Initially in the absence of calcium the probe shows very weak emission in aqueous solution while in the presence of calcium the formation of fibrillar takes place resulting in the enhancement in fluorescence, as shown in Figure 5 [40]. 

In another report, Wang and coworkers designed AIE active polyarylene ether nitrile fluorescent nanoparticles acting as an excellent probe for intracellular imaging. The synthesized nanosphere showed excellent biocompatibility in presence of calcium with a 21% quantum yield [41]. Moreover, another TPE-based AIE active fluorescent probe was synthesized by Zhang’s group for sensing Ca^2+^ using the criteria of AIE behavior. The probe consisted of a TPE moiety with a bidentate pyridine carboxylate unit. The probe showed a “turn-on” fluorescence response to Ca^2+^ with a lower detection limit of 51.2 nM. Importantly, the probe can also be recycled by addition of ethylenediamine tetra-acetic acid (EDTA) [42] and further reused. Ishiwari’s group demonstrated use of AIE-active TPE-based solid state gel sensor fluorescent molecule for extracellular Ca^2+^ imaging. This gel consisted of polyacrylic acid (PAA) functionalized with TPE produces PAA-TPE, in which TPE acted as a pendant with an AIE-active property. Interestingly, crosslinking with gel (g-PAA-TPE) showed good selectivity towards Ca^2+^ [43]. Figure 6 shows various AIE active fluorophores based on TPE for detection of Ca^2+^.

In the human body it is well known that after iron, zinc is the second most common transition metal ion. Along with Na^+^, K^+^ and Ca^2+^, Zn^2+^, plays a very important role in various biological and physiological functions such as the nervous system, gene transcription, immune functions, and mammalian reproduction. In fact, one of the most important applications is that the Zn^2+^ acts as an excellent catalytic cofactor and structural center in various enzymes, DNA, and proteins. Therefore, the need for studying and monitoring the Zn^2+^ is a major aspect in sensing applications. By taking advantage of AIE-activity of molecules, Sun and coworkers designed a TPE-based “turn-on” fluorescent probe for detection of Zn^2+^ ion in an aqueous medium. In this, co-ordination of Zn^2+^ to -N(CH_2_COO^−^)_2_ and intermolecular coordination of Zn^2+^ lead to aggregation resulting in an enhancement in fluorescence [44]. In another report, Wei and group synthesized AIE active multifunctional metal-organic vesicles with triarylamine carboxylate (TPA-1) for specific detection of Zn^2+^ ions. This fluorescent probe can be employed for biological cell imaging and drug delivery systems [45]. Another multifunctional TPE-based AIE active fluorescent probe was designed by Tang and coworkers for selective and sensitive detection of Zn^2+^ and Hg^2+^ ion with 1.24 × 10^−6^ molL^−1^ and 2.55 × 10^−9^ mol L^−1^, respectively. The AIE activity of the fluorescent probe was analyzed using the THF-water fraction [46]. More recently, Maity et al. developed an antipyrine fluorescent probe 4-[(2-hydroxy-3-methoxy-benzylidene)-amino-1,5-dimethyl-2-phenyl-1,2-dihydro-pyrazole-3-one (OVAP) for selective detection of Al^3+^ and Zn^2+^ detection [47]. Diana and Panunzi in their review described several AIE active fluorescent probes for sensing of Zn^2+^ ions [48]. Moreover, Sun et al. developed a multifunctional AIE active Schiff base fluorescent probe (TPESB) combined with AIE and ESIPT. TPESB was employed for dual-channel sensing of Zn^2+^ with high selectivity and sensitivity exhibiting a low limit of detection of 38.9 nM. Thus due to its low cytotoxicity, the probe is applied for sensing of Zn^2+^ in live cells [49]. In this work, He et al. designed a “turn-on” fluorescent probe with TPE as AIE active fluorophore combined with the peptide chain. The self-assembled complex was formed between the Zn^2+^ and three histidine residues. The probe showed strong fluorescence emission in presence of Zn^2+^. The observed limit of detection was 18.56 nM and it can be employed for biological cell imaging and intracellular detection of Zn^2+^ having low cytotoxicity and good stability [50]. Figure 7 shows the various fluorophores used for detection of Zn^2+^ ion. 

There are several non-essential metal ions or heavy metal ions including anions that are very well known for their toxicity to human health causing severe diseases. The major factor responsible for the entry of these toxic metal ions into the environment is due to industrialization, modern agricultural practices, and the release of untreated waste directly into water bodies resulting in contamination and destruction of natural resources indirectly affecting human health [51,52,53,54]. Different toxic ions that have toxic effect are Al^3+^, Cr^3+^, Cd^2+^, Co^2+^, Mg^2+^, Hg^2+^, Sb^3+^, CN^−^, F^−^, Br^−^, Cl^−^, As^3+^, As^5+^, Cu^2+^. Therefore, there are several analytical tools have been employed for the detection of this metal ion and anion in the system such as atomic absorption spectroscopy, inductively coupled plasma mass spectrometry, and colorimetric methods. However, fluorescent chemosensors have gained a lot of advantages over all these methods for several reasons such as fast, cost-effective, simple and easy to handle, naked-eye detection, and low cost of instrumentation. However, few organic fluorescents suffer from aggregation-caused quenching thanks to the AIE phenomenon developed to overcome the difficulties faced due to ACQ. Therefore, here different AIE active fluorescent probes (Figure 8) for detection of toxic metal ions and anions are described [55,56].

Beglan et al. synthesized a derivative of TPE functionalized with cysteine as the fluorescent AIE active dye for the detection of Arsenic (As^3+^) in aqueous media [57]. Further investigation of the probe revealed that the thiol group of cysteine acts as a binding site for arsenic via the As–S bond. The three binding units of cysteine combines to sense As^3+^ to give a symmetrical As–(Cys-TPE)_3_ complex. Wen and group synthesized a novel AIE triphenylamine fluorophore used for sensing of Hg^2+^ and CN^−^. The probe showed excellent AIE active behavior and has potential application in biological cell imaging. However, the probe can be recycled and reused without any loss [58]. In another report, a highly selective and sensitive pyrene-based AIE active ratiometrics “turn-on” fluorescent probe (pyrene-DT) was designed by Ma co-workers for detection of Hg^2+^ in aqueous media. The probe has good practical applicability in the preparation of test strips and detection of Hg^2+^ in a water sample [59]. Our group recently described different AIE active luminogens that are designed for selective and sensitive detection of various toxic metal ions [60]. Elemental copper plays a very important role in various physiological processes in the environment. However, excess and deficiency of copper may lead to various neurological disorders, mostly kidney and liver damage. More recently, our group synthesized TPE-based AIE active comprising a thiophenylbipyridine receptor for selective and sensitive detection of Cu^2+^. The probe showed a very low limit of detection up to 7.93 nM and the probe was further used for test strip preparation, which is one of the best advantages for practical applicability [61]. In a similar direction, Jiang and coworkers designed double detecting hydrazono-bis-tetraphenylethylene (Bis-TPE) based AIE active fluorescent probe for Cu^2+^ and Al^3+^. Here, the probe showed quenching of fluorescence to Cu^2+^ and red-orange fluorescence for Zn^2+^ with 1:1 stoichiometry. However, the strong fluorescence of Bis-TPE+Cu^2+^ can be recovered by adding adenosine triphosphate (ATP) giving “turn-on” fluorescence similar to Bis-TPE+Zn^2+^ that can be quenched by adding Cu^2+^ which is then further recovered by adding ATP. The probe Bis-TPE can be employed for test strips sensing and biological cell imaging [62]. Zhang and colleagues synthesized click triazole bridged cyclodextrin (CD) based AIE active molecules for sensing Cd^2+^. The probe possesses a good AIE active property and shows a “turn-on” fluorescence response towards Cd^2+^ ions. The limit of detection for the probe was found to be 0.01 μM. Interference studies in the presence of only Cd^2+^ showed a good response [63].

Similar to cations, anions sensing has been in more demand due to its wide application in a biological and chemical processes, even though the anions are also considered detrimental and toxic to the environment and human health. The fluorescence method has proved to be the best approach for detection, monitoring, and remediation of anions such as SO_4_^2−^, CN^−^, F^−^, NO_3_^−^, Cl^−^, Br^−^. Most recently, our group has designed and synthesized a TPE-based AIE active fluorescent probe which is highly selective and sensitive towards cyanide. It can be seen clearly that the probe can be efficiently utilized for naked-eye detection Figure 9 test strips, and importantly the fluorescent probe, were used for the detection of CN^−^ ion in biological food samples. Moreover, the probe was utilized for a biological cell imaging application [64].

Nhein and coworkers designed a novel amphiphilic AIE active polymeric fluorescent material, poly(NIPAM-co-TPE-SP) prepared by NIPAM as a hydrophilic unit and tetraphenylethylene-spiropyran monomer (TPE-SP) employed for detection of cyanide in aqueous media. The unique property of this fluorescent probe is that in the presence of UV light the closed spiropyran (non-emissive) poly(NIPAM-co-TPE-SP) opens to give merocyanine (MC) (poly(NIPAM-co-TPE-MC) in an aqueous solution [65]. In addition, our group has synthesized another AIE-active TPE-based cyclic urea-based receptor for selective detection of F^−^ ions. The probe showed good selectivity in the presence of other anions such as Cl^−^, Br^−^, I^−^, HCO^3−^ CO3-, NO_2_^−^, NO_3_^−^, SO_3_^2−^, SO_4_^2−,^ AcO^−^, S^2−^, ClO_4_^−^, HPO_4_^−^, CN^−^. Absorption and fluorescence studies revealed that only F^−^ ion showed excellent selectivity towards the cyclic urea probe [66]. Similarly, using tetraphenylethylene as a moiety to monitor optical as well as calorimetric changes, Anuradha et al. synthesized an amino-functionalized fluorescent probe i.e., tetraamino-TPE (TA-TPE) for the selective detection of nitrite ion in aqueous media. The probe showed a “turn-on” fluorescence response which was considered an advantage over other receptors [67]. In another example, TPE containing AIE active metal-organic supramolecular nanobelt was developed by Li et al., and in their work the TPE was functionalized with terpyridine moiety to give tetrapodal TPE-terpyridine ligand which self assembles to give a metal-organic nanobelt which shows high selectivity towards S^2−^ [68]. AIE active luminogens did not only limit its application to sensing of cations and anions but also showed wide application in detection of pesticides [69], explosives [70,71,72], biomolecules [73,74], pathogens [75], food additives [76] and so on.

## 3. AIE Active Molecules for Biological Cell Imaging

The various fundamental processes taking place in life can be very well studied and understood which can be achieved effortlessly using fluorescence technique. Fluorescence microscopy is an important tool that has now become most advantageous and launched in biological research for advance and a better understanding of various intracellular processes and dynamics at a cellular level. The field of fluorescence imaging is actively developing as this fluorescent tool plays a major role in monitoring the different analytes and helps in studying the biological events occurring in the intracellular environment. Fluorescence microscopy is one of the best techniques to have been used in the last few decades. Nowadays, there is great progress in instrumentation techniques which makes fluorescence imaging simple and also overcomes the drawbacks and challenges suffered in the past. Doing so, various advantages may be tackled such as minimal photodamage, high resolution, and penetration deep into tissues. Nevertheless, in fluorescence microscopy, two-photon fluorescence imaging has a greater advantage over one-photon fluorescence imaging [77].

In this regard, Li and co-workers designed organic far-red/near-infrared AIE active TPE substituted dots such as TPE-TPAFN and TPA-FN with excellent utility in long-term cellular tracing. Due to its excellent characteristics of high emission efficiency, large absorptivity, excellent biocompatibility, and also its photobleaching resistance, the fluorescent probe is a good candidate for in vitro as well as in vivo cellular tracing. The matrix used for fabrication is the mixture of polyethylene glycol (PEG) and lipid-PEG-NH_2_. Interestingly, the Tat (transactivator of transpiration)-AIE dots demonstrated excellent imaging properties [78,79]. 

Wang et al. used “turn-on” fluorescence based on AIE active TPE in conjugation with chitosan to produce chitosan-TPE bioconjugate which can be used as “turn-on” fluorescence for long-term cellular tracing, which is very important for monitoring biological and therapeutic processes. This bioconjugate exhibits excellent AIE active property followed by internalization of aggregate take place in HeLa cells for further biological cell imaging. The HeLa culture was used and incubated for 24 h and the fluorescence images were recorded, and the cell imaging process was carried out until the 15th passage, Figure 10 [80].

In the fluorescent materials, far-red/near-infrared (FR/NIR) emissions higher than 640 nm are regarded as an excellent and promising candidate for fluorescence imaging. Using similar criteria, the Qin group synthesized selenium containing FR/NIR AIE active luminogen (TTSe dots) which is very rare; however, there are very few reports available on selenium-containing fluorescent probes for bioimaging application. The synthesized probe has excellent applications in imaging of the sentinel lymph node (SLN) mapping and finally in tumor imaging. In addition, this fluorescent luminogen is utilized for imaging the brain blood vessels of mice using a high-resolution two-photon imaging technique as shown in Figure 11. Hence this highly emissive selenium-containing TPE-based fluorescent luminogen is used for in vivo biological applications [81]. 

Similarly, Gao and coworkers developed AIE functionalized with Tat-peptide (AIE-Tat NPs) PITBT-TPE for tracing bone marrow mesenchymal stem cells (BMSCs). The fluorogens consisted of a mixture of DSPE-PEG_2000_ and DSPE-PEG_2000_ maleimide matrix. Furthermore, the obtained nanoparticle was modified with cysteine Tat-peptide (RKKRRQRRRC). The fluorescence quantum yield of the compound was found to be 23.5% with a lifetime of 5.37ns. Thus, the red emission made the AIE-Tat NPs a promising candidate for bioimaging of BMSCs cells; however, intense red emissions were observed in a confocal microscope. The results indicated 100% labeling efficiency and the concentration of AIE Tat does not affect the cell viability of BMCs cells [82]. Huang et al. reported AIE-active fluorescent luminogen for detecting and intracellular imaging of ClO^−^ in cells. The TPE unit being hydrophobic forms aggregate nanoparticles into micelle that emits red fluorescence. Therefore, the probe can be utilized for endogenous ClO^−^ detection in live cells. To study the endogenous imaging of ClO^−^, the zebrafish was used as a sample because of their excellent transparency at the embryonic and larval stages. Zebrafishes were incubated with the fluorescent compound for 60min, and green fluorescence was observed however the green fluorescence enhanced with increasing incubation time. This result reveals that the probe can be employed well for in vivo ClO^−^ detection and imaging applications [83]. The Ma group synthesised AIE-active TPE polymer cross liked using N-isopropylacrylamide fluorescent material for long-term cellular tracing. This fluorescent polymer was designed with hydrophilic N-isoprpylacrylamide polymer and hydrophobic TPE subunits with cross-linkers 4,4′-(2,2-dibromoethene-1,1-diyl)bis(vinylbenzene)DDBV. The synthesized TPE-PNIPAM (P6) had good compatibility and long-term imaging properties. In addition, the probe showed temperature-responsive behavior with fluorescence change. Its excellent biocompatibility makes the fluorescent probe a highly promising material for a biological imaging application. Thus, it was observed that the probe was utilized for examination of living A549 human lung adenocarcinoma cells as shown in Figure 12, in which the deep blue color of the cytoplasmic cells suggests the P6 molecule aggregates and internalizes deep into the cells. The P6 molecule shows good AIE activity, low toxicity, and leakage-free staining. Hence it can be concluded that the P6 molecule acts as an excellent fluorescent marker for a biological long-term cell imaging application [84]. 

Moreover, the Wan group prepared AIE-active fluorescent polymeric nanoparticle using the polysaccharides, as many of the fluorescent polymeric nanoparticles (FPNs) have increasing demand in theranostic application especially in drug delivery systems in ttreatment and diagnosis. In Wan group work, they utilized cost-effective naturally occurring polymer oxidized sodium alginate (OSA) obtained from marine seaweeds. The designed fluorescent AIE active OSA-Phe-OSA FPNs show red fluorescence with excellent photostability and biocompatibility. The biological cell imaging studies were conducted on L929 cells using OSA-Phe-OSA FPNs. They showed the cellular uptake of the fluorescent probe was effective with excellent biocompatibility. The strong fluorescence was observed after 3 h of incubation which suggests that the OSA-Phe-OSA has excellent staining performance. They concluded that these OSA-Phe-OSA AIE-active nanoparticles can be used in future for controlled drug delivery of cisplatin [85]. 

Lipid droplets are the powerhouse of proteins and lipids responsible for various biological processes and diseases, for example virus infection, cancer, obesity and so on. Several approaches are being used earlier for visualization such as color staining, Raman microscopy, transmission electron microscopy. Amongst all these techniques near-infrared and two-photon excited fluorescence (TPEF) are more advantageous over other techniques due to their high-resolution imaging and low photodamage. Therefore, Gao et al. introduced AIE active triphenylamine (TPA) and indane-1,3-dione optical material. The compound showed good AIE activity and TICT effect hence can be used for lipid droplet specific imaging and thus making it easily available for biological applications. The biological cell imaging on HCC827 and A549 cells were carried out which suggested that upon incubation for 15 min the fluorescent probe IND-TPA shows excellent cellular uptake property [86]. Another AIE active TPA-based dihydro-2-azafluorenones was used for lipid droplet-specific live-cell imaging [87]. 

Fluorescence microscopy is considered an important tool in the imaging of tissue. Due to optical diffraction limits, the spatial resolution is restricted up to nearly 200 nm which keeps the imaging unclear. After several decades numerous high-resolution methods were discovered. In this direction, Li and co-workers synthesized AIE active nanoparticle (NPs) of 2,3-Bis(4-(phenyl(4-(1,2,2-triphenylvinyl)phenyl)amino)phenyl)fumaronitrile (TTF), which was fabricated with colloidal mesoporous silica to form AIE-active NPs which are used for cell imaging by utilizing the stimulated emission depletion nanoscopy (similar to confocal microscopy) which is a super-resolution imaging technique as it gives a resolution of <50 nm that is more advantageous over other fluorescence imaging tool. Moreover, utilizing the long-term stimulated emission depletion (STED) nanoscopy TTF@SiO2 NP was investigated for imaging cancerous HeLa cells. Figure 13 illustrated that the images obtained under a confocal microscope and STED showed two different results; it clearly shows that the images obtained by STED are very high-resolution images compared to the confocal microscope. Therefore, fluorescent TTF@SiO_2_ NPs are beneficial for long-term super-resolution bioimaging applications in future technology developments [88].

Supramolecular interactions play a very important role and have been widely used for the fabrication of various fluorescent assembled nano- and micro-superstructures. In this regard, the Xu group fabricated AIE active fluorescent organic nanoparticles (FONs). The FONs were synthesized having an AIE active property exhibiting red emission, however they were fabricated through the supramolecular interaction between β-cyclodextrin (β-CD) and adamantine AIE active dye. Due to its AIE activity of Ph-Ad/β-CD and strong red emission, the FONs can be used for an imaging application. To check its biocompatibility cell viability studies were carried out using a CCK-8 assay. The 100 % cell viability was observed for A549 upon incubation with FON of cells for 24 h. Cell uptake studies for the FONs were evaluated and confirmed that due to low cytotoxicity the Ph-Ad/β-CD, FONs acts as an outstanding platform and thus can be utilized for further biological and biomedical application [89]. Another cyclodextrin-TPE substituted AIE active pseudorotaxAne was synthesized by Liow and group. The prepared luminogen consists of tetraphenylethylene conjugated with poly(ethylene glycol) (TPE-PEG2) as a guest species and α-cyclodextrin(α-CD) as a host species. The molecule was investigated for cellular uptake and examined for cytocompatibility by internalization of TPE-PEG2 for A549 cells. The confocal images showed that the treated cell exhibited blue fluorescence to the cytoplasm upon incubation with TPE-PEG2, however, upon addition of α-CD to TPE-PEG2 the blue fluorescence was enhanced significantly. They concluded that the blue fluorescence was only observed in the cytoplasm rather than the nucleus Figure 14 [90]. 

Guan and coworkers synthesized AIE active non-conjugated polymer dots (Pdots) by fabricating amphiphilic PNIPAM with Eu (III) complex which self assembles in aqueous media as a four-arm star polymer TPE-tetraPNIPAM-Eu (III). The developed fluorescent material is employed for cancer cell imaging. The Pdots show dual fluorescence emission properties from TPE hydrophobic moiety and hydrophilic PNIPAM-Eu (III). Due to its dual emission property, blue light at 630 nm and red light at 395nm make it a promising candidate for cell imaging of HeLa cells, A549, and HepG2 cells upon incubation with AIE Pdots. The cytotoxicity studies revealed 90% cell viability Figure 15 [91]. 

Zhuang et al. synthesized AIE-active redox and pH-responsive polymeric micelles mPEG-P(TPE-co-AEMA). This synthesized polymeric micelle has excellent biocompatibility, it acts as a prominent nanocarrier for drug delivery system. The cellular imaging was carried out using a mPEG-P(TPE-co-AEMA) micelle for 4T1 cells and HeLa cells. Both the cells showed 100% cell viability indicating that the micelle has excellent biocompatibility. The confocal laser scanning microscopy (CLSM) images of 4T1 cells as represented in Figure 16 [92]. 

Zhang and coworkers designed amphiphilic TPE-based pyridinium salt with yellow emissive (TPE-MEM) with a unique AIE active phenomenon. The probe has good water solubility, biocompatibility, and cell membrane specificity. The cytotoxicity study revealed that the cells were viable to 20 μM with low toxicity. The cell imaging studies were undertaken by using cervical cancer HeLa cells which were incubated with TPE-MEM at room temperature with 5 μM. The excess of PBS buffer was removed by washing and the images were recorded as shown in Figure 17 [93]. 

Feng et al. demonstrated a position-dependent substituent effect on another TPE-based AIE active fluorescent probe functionalized with pyridine moiety (TPE-o-Py) which displays high selectivity and sensitivity towards the Fe(III) ion. In addition, in vitro cellular imaging and selectivity of Fe(III) was carried and this fluorescent probe shows pronounced red-shift and is applicable in biological cell imaging. The HeLa and MCF7 cells were incubated with the fluorescent probe, and images were captured under confocal fluorescence microscopy. The HeLa cells exhibited blue emission upon incubation with TPE-*o*-Py, but as the Fe^3+^ was added the blue, as well as red, emission was illustrated. Upon incremental addition of Fe(III) red fluorescence intensity increases rapidly. Its excellent biocompatibility reveals that TPE-*o*-Py acts as an excellent boilable for Fe(III) in both Hela and MCF7 cells [94]. In another report, Li and coworkers synthesized the alkylated functionalized pyridinium TPE system (TPEPy-1 to TPEPy-4). The effect of substitutes chain length, optical properties, and its further application in the various biological field was investigated. Moreover, the alkylated chain influences the optical properties of TPEPy-1 that found to be a promising “turn on” fluorescence candidate for detection of NO_3_^−^ and ClO_4_^−^. The fluorophores were used for mitochondrial targeting. To study the effect of alkyl substituent the HeLa cells and HEK-293T were selected for cytotoxicity, eventually investigated by CLSM and subsequently showing that TPEPy-1 and TPEPy-2 get accumulated and represent the yellow fluorescence; however, weak fluorescence was observed for TPEPy-3 and negligible fluorescence for TPEPy-4 fluorogens. Thus, they have concluded that the TPEPy-1 and TPEPy-2 due to suitable hydrophobicity accumulate well in the cell membrane while a very strong hydrophobic nature makes it impossible for fluorogens to penetrate deep in the tumor as well as the normal cells Figure 18 [95].

However, it is critically important for a cell viability study of therapeutic efficacy in biomedical applications. Therefore, Hu and co-workers designed and synthesized AIE active polymeric material with TPE and a negatively charged side chain of OEGs P(TPE-OEG) to study the viability of cells. AIE-active polymer demonstrates good tracking ability and excellent biocompatibility [96]. The Lin group developed AIE-active “turn-on” probe for alkaline phosphate detection by introducing an electron-withdrawing group and recognizing phosphate species. The probe is highly sensitive to alkaline phosphate (ALP) and has been successfully employed for imaging ALP in live cells [97]. In 2019, Chen and coworkers demonstrated several fluorophores with tetraphenylethylene dye and dansyl, naphthalimide, 4-nitro-1,2,3-benzoxadiazole (NBD), borondipyrromethane (BODIPY), hemicyanine fluorophore in visible to near-infrared regions exhibiting AIE active effect. These fluorophores TPE-NIR, TPE-Blue, TPE-Crimson, TPE-Orange, and TPE-Red have been further explored in biological cell imaging, Figure 19 [98]. The AIE active TPE-containing fluorescent dyes have been then applied for the bioimaging in living cells, which covered the broad emission region. The dyes solution was prepared in phosphate buffer of pH=7.4 with 0.1% of DMSO. After incubation of HeLa cells the fluorescence images were taken in confocal microscope as shown in Figure 19, bright emission was detected in living cells using dyes TPE-Blue, TPE-Orange, TPE-Red, TPE-Crimson, and TPE-NIR, represented in different colours.

By using the specific sugar–protein interaction strategy, Wang and coworkers developed a diketopyrrolopyrrole (DPPM) AIE-active probe with two mannose groups and diketopyrrolopyrole (DPP) core. They further studied the effect of an increasing number of sugar groups on the DPPM two DPPF-M (with four mannose units) and DPPS-M (with six mannose units). All three DPPM, DPPF-M, and DPPS-M fluorophores showed AIE activity which was further employed for the detection of lectin. The fluorescence of the DPP unit increased upon the addition of lectin. Interestingly, the probe showed low cytotoxicity towards MCF-7 and MDA-MB cells and had greater capacity for recognition of sugar ligands and glycoproteins cellular imaging of cancer cells which overexpressed mannose receptors through a specific sugar–protein interaction. In which, DPPS-G was used as a reference without any mannose unit for comparative cell imaging study with DPPF-M and DPPS-M probe. The laser scanning confocal microscopy (LSCM) results show that upon incubation of cells with DPPS-M and DPPF-M, units showed red fluorescence to cancer cells while no fluorescence to normal cells. In addition, they noticed that cells upon incubation with DPPM had very weak red fluorescence, and with DPPS-G showed no red fluorescence. This indicated that only the mannose functionalized DPP-based AIE-active compounds with a greater number of mannose groups comprising a greater number of binding sites are responsible for cellular uptake resulting in its utilization in a biological cell imaging application [99].

Yan et al. synthesized di(2-picolyl)amine(DPA) which was highly selective towards the Zn(II) metal ion. The modified DPA to form 3-amino-9-ethyl carbazole with salicylaldehyde was prominently utilized for detecting the Zn (II) ion in aqueous media. The probe applied for biological cell imaging. Initially, cytotoxicity studies were undertaken to study the viability which revealed low toxicity with 97.7% viability. Thus the probe was used for recognizing Zn(II) ion in HeLa cells [100]. Wang’s group synthesized an AIE-active fluorescent chemosensor based on triphenylamine-based Schiffs bases C1 and C3 with reference to C2 and C4 (Figure 20). The synthesized C1 and C3 showed good AIE activity. The synthesized AIE probe was utilized for the detection of hydrazine N_2_H_4_∙H_2_O in living cells. The HeLa cells were incubated with probes C1 and C2. The results indicated that upon incubation with hydrazine there was the disappearance of fluorescence under a fluorescence microscope, suggesting that the presence of hydrazine makes the probe C1 and C2 applicable for the detection of hydrazine in live cells [101]. 

Tarai and co-workers synthesized two cyano functionalized ICT and AIE active probes for cytotoxicity and cell imaging applications. The molecules with substitution effect on AIE-1 and AIE-2 were investigated. The AIE-1 without substitution showed yellow color with weak charge transfer whereas AIE-2 showed strong charge-transfer absorption with yellowish-orange color. In both AIE-1 and AIE -2, indole moiety acts as an electron donor and the cyan-functionalized group acts as an electron acceptor unit. Furthermore, both the probes were investigated for cell imaging by using the HeLa cells as the model system for cellular uptake. The toxicity studies revealed good cell viability to 90% even after increasing cell incubation period. Furthermore, the HeLa cells were incubated with AIE-1 and AIE-2 for 24 h and at 37 °C. The images were captured by a confocal microscope under bright and greenfield as illustrated in Figure 21 [102].

Thermoresponsive polymeric AIE active (1-ethenyl-4-(1,2,2-triphenylmethanol)-benzene-b-N-isopropyl acrylamide micelles were developed very recently by Ma and co-worker by reversible addition-fragmentation chain transfer polymerization of 1-ethenyl-4-(1,2,2-triphenylethenyl)-benzene (TPEE) and N-isopropylacrylamide (NIPAM). The polymer PTPEE-Pns possesses a good AIE-active property. As the polymeric micelle showed excellent stability and bright fluorescent emission, the micelle was further explored for cell imaging. Before cell imaging, the micelle was subjected to a cytotoxicity study which revealed low cytotoxicity with 95% cell viability. The micelle exhibited blue fluorescence upon incubation and blue fluorescence was observed in the cytoplasm rather than the nucleus. Thus, it was considered a promising candidate for cellular imaging [103]. 

Various biological and cell functions are accompanied via the plasma membrane which is the important building block of a cell. In this regard, Sayed et al. reported photostable AIE active probe for a plasma membrane imaging study comprising of tetraphenylethylene naphthalimide (TPE-NIM^+^). The TPE-NIM^+^ was used for staining of different cell lines acting as an excellent fluorescent marker [104]. In another example, the polymeric micelle was fabricated into mesoporous silica hollow nanosphere (MSHN) by Kumar and co-workers. The polymeric material was synthesized by using AIE-active TPE and triphenylamine by reacting with polymer poly (N, N-diphenyl-4-(4-(1,2,2-triphenylvinyl) styryl) aniline) (PTPA) comprising the D-п-A system which is further incorporated into a mesoporous silica hollow sphere. With these PTPA-loaded MSHNs, conjugated polymeric nanoparticles were further investigated for cell imaging application by fluorescent microscopy via internalization of the probe into the Huh-7 cells, Figure 22 [105].

## 4. AIE Molecules for Drug Delivery Systems

Nowadays diseases and disorders are increasing at an alarming rate and therefore proper treatment and dosage of medicine/drugs have become one of the major aspects for treating disease and improving human health. Drug delivery is defined as the process of delivering or releasing a drug/bioactive agent to a targeted site at a specific rate. Drug delivery is an interdisciplinary field and has gained a lot of attention from many researchers in pharmaceuticals, medical, doctors, and industries as it plays a major role in delivering medicine or a drug to its therapeutic site. Cancer is the major cause of death and at present chemotherapy is frequently used for treatment, but currently there are several limitations faced such as lack of tumor specificity, few drugs due to poor solubility, higher toxicity, aggregation due to lack of solubility; nevertheless, few drugs degrade in vivo before reaching the target site, and sometimes drugs deliver to non-specific sites which make biotechnology very challenging. Thus, biotechnology has growing demand for engineering physics, chemistry, biology, and medicine [106]. At present, developments in biotechnology and various fields are designing and developing a new class of drugs that play a crucial role in delivering to the targeted site, which is turning out to be a new advance in clinical diagnosis. Moreover, the field of drug delivery is now developing rapidly as many researchers from different research fields have combined to overcome the challenges faced during drug transport. 

In the last few decades, several nanocarrier molecules were used as drug delivery vehicles and considered as an excellent platform to carry different nucleic acids, antibodies, photosensitizers, imaging agents, and anticancer drugs that help in diagnosis and therapeutics [107]. There are several functionalized nanomaterials, including nanoparticles [108,109,110], yoctowells models [111,112,113,114], Cathepsin [115], polymeric nanoparticle [116] porphyrin-based nanomaterials [117], quantum dot clusters [118], mesoporous silica particle (mSiO_2_) [119,120,121] magnetic nanoparticles [122], functionalized miscelles [123], cyclodextrins [124,125] and so on. Besides all these nanocarriers, there are functionalized AIEgens which nowadays are extensively used in drug delivery systems [126]. However, many traditional nanocarriers are invisible making it impossible to undertake intracellular trafficking which is the limitation on the drug delivery system. Therefore, a nano-drug delivery system with fluorescence imaging was developed and needed to be promoted [126,127]. Many fluorophores suffer from ACQ which hampers fluorescence; therefore, the novel AIE active fluorophore made it possible to overcome the ACQ and was found to be a very promising application in biomedical application. The AIEgens are not only used for sensing, bioimaging, and diagnosis purposes but the development of AIEgen-based systems has shown its tremendous application in diseases theranostics, such as image-guided chemotherapy, photodynamic therapy (PDT), gene delivery, photothermal therapy (PTT), or a combination of two or more methods [128,129]. The development in drug release mechanism with the AIEgens is visualized in Figure 23 [130].

In this regards, Li et al. prepared TPE functionalized organophosphonic acid, (4,4′-(1,2-diphenylethene-1,2-diyl)bis(4,1phenylene))-bis(methylene diphosphonic acid (PATPE) with excellent AIE activity [131]. The molecule was further incorporated into HAp, which is a three-dimensional structure via P–O–Ca covalent bonds forming an ellipsoidal hollow nano capsule. The following luminescent molecule was fabricated with hydroxyapatite to give a hollow mesoporous nano capsule emitting strong blue light. The molecule was used for drug delivery using ibuprofen (IBU) as the drug moiety. The drug release is well studied by its change in fluorescence intensity. The fluorescence intensity change observed during loading and release of IBU was monitored by a fluorescence study [131]. Similar study Soon after that, Xue et al. developed another TPE-based TPE/DOX nanoparticles (TD NPs) for cancer therapy [132]. Doxorubicin was added to TPE-based NPs in different proportions (5%, 10%, and 15%) The TD with doxorubicin was successfully fabricated and it was confirmed by fluorescence studies that the TD NPs exhibited fluorescence of both TPE as well as of DOX indicating the formation of TD NPs which show weak fluorescence. Thus, it was further concluded that DOX can be recovered once it has been released from TD NPs. However, the TD NPs are pH-responsive so that the DOX is released in the lysosomes at low pH 5.0. Initially, TPE-COOH is negatively charged while DOX is positively charged which shows the interaction at neutral pH 7.4 to form a nanoparticle. It was observed that the FL of both TPE and DOX reduces, but a significant change was observed at pH 5.0 as the DOX is released in lysosomes thus making it reversible. Furthermore, TD NPs were dispersed in acetate buffer at pH 5.0. The fluorescence of DOX slowly returns, which confirms the detaching nature of DOX from the parent TD NPs. This confirms that TD NPs and DOX are pH-responsive; as the pH of solution changes from 7.4 to 5.0 FL its intensity increases which signify that the DOX is released from TPE NPs. Thus, it can be concluded that DOX can be released to the lysosomal cell at low internal pH [132]. In addition, Zhang and co-workers developed a new drug delivery system comprising TPE fabricated with micelle exhibiting excellent AIE activity. This nanocarrier with switching ‘ON’- and ‘OFF’-active mode is controlled by assembly and disassembly of the micelles with AIE making it more permissible for high-quality imaging. This DDS nanocarrier was further analyzed for doxorubicin (DOX) release and cellular imaging. It was observed that DOX-loaded micelles (TPED) had good efficiency compared to free DOX. Upon further investigation for drug delivery capacity of the DOX-loaded TPE, it was observed that at low pH 5.0 in lysosomes DOX can be released at low internal pH [133]. Polymeric material coordinated to TPE also play a significant role in drug delivery. Wang et al. synthesized TPE zirconium-based nanoscale coordination polymer (TPE-NCPs) exhibiting AIE behavior. Herein, tetrakis(4-carboxyphenyl) ethylene acid was incorporated into zirconium-based NCPs. Furthermore, the TPE-NCPs were utilized for drug delivery by taking curcumin (Cur) as an anticancer drug to study its drug loading and drug release mechanism. However, for cellular uptake, it is essential to have NCPs with an approximate size of less than 200 nm. Therefore, the NCP with the appropriate size was synthesized at 90 °C by using acetic acid (HAc) as a modulator. As the amount of modulator (HAc) increases gradually the size of the NCPs also increases (20 μL, 50 μL, 100 μL, 150 μL and 300 μL of HAc). It was noted that NCP-1-20, NCP-1-50, and NCP-100 of less than 50 nm size and NCP-1-150 form larger sized nanoparticles of 70 nm upon addition of 150 μL of the modulator. Therefore, NCP-1-150 was used for a drug release and drug-loading mechanism. Initially, the NCP-1-150 was mixed with Cur in methanol with constant stirring to room temperature for 12 h. The successful load of the Cur drug was confirmed by the UV absorption study. The efficiency of the loaded drug in NCP-1-150 was found to be 65%. The drug release was monitored at 37 °C in saline phosphate buffer at pH 7.4 and 5.0. Thus, that a stable and continuous release of Cur drug from Cur@NCP-1-150 was observed. Around 53.7% of Cur is released in an acidic buffer system while 40.4% at a neutral PBS system within 96 h. Drug release was further confirmed by the PXRD pattern of the used nano-drug system. After the release of the drug the structure of the nanocarrier remains stable and can be applied as an excellent platform for the drug loading and release mechanism [134]. Chen and co-workers [135] developed pH-responsive polymeric micelles with zwitterionic copolymer poly(2-methacryloyloxyethylphosphorylcholine-co-2-(4-formyl-phenoxy)ethyl methacrylate)(poly(MPC-co-FPEMA)) through the AIE active molecule by RAFT polymerization method which converts it to PMPC-*hyd*-TPE via conjugation with TPE consisting of acid cleavable hydrazone bonds. Further micelle was loaded with DOX forming the hydrophobic interaction, as shown in Figure 24. At low pH of 5.0, the TPE moiety linked to DOX becomes cleaved leaving behind free micelle. The release of drug DOX from TPE-DOX was studied in the pH range of 7.4 t 5.0. It was observed that at physiological pH, about 20% of DOX was released compared to pH 5.0 where about 70% DOX was free from the parent PC-*hyd*-TPE-DOX micelle in 24 h. The self-assembled TPE micelle disassembled itself under acid conditions through bond cleavage as the DOX release increased. Thus, this suggests that the micelle acts as an excellent candidature for drug release under an acidic condition in tumor cells. In addition, the micelle was utilized for cell imaging.

In vivo drug release monitoring by using AIE-active thermogelling polymer was carried out by Liow et al. [136] The self-indicating drug delivery system was developed for long-term monitoring with the following characteristics having an injectable non-invasive system that can monitor the bulk state of materials without any leaching of fluorescent dye and further encapsulation of the drug and protein. This thermogelling polymer (EPT) was synthesized by functionalizing TPE moiety with poly (PEG/PPG/TPE urethane) as shown in Figure 25. At CMC, the universe self-assembles itself in micelles containing PPG and TPE as a hydrophobic moiety and PEG as a hydrophilic chain, all forming together the micelle cluster. The developed PEG, PPG, and TPE is dependent on DOX concentration and temperature. This AIE-active thermogel is more advantageous than other DDS as the thermogel matrix can reveal the drug release status. The EPT was found to be non-toxic and showed high efficacy in reducing the tumor size. Thus, the synthesized gel is a self-indicator with excellent biocompatibility, thus being a good candidate for in vivo detection of drug concentration and a drug delivery mechanism.

Hyaluronic acid (HA) is a naturally occurring polysaccharide found mostly in the extracellular matrix, epithelial connective tissue widely used in biomedical applications especially in drug delivery, tissue engineering, and molecular imaging. HA is considered as a prodrug nanocarrier in combination with DOX, paclitaxel, and mitomycin Cor siRNA for cancer therapy. In this regard, Wang’s group modified HA with phosphorylcholine in conjugation with AIE active moiety and DOX system. Usually, the HA forms micelles which are internalized into cancer cells and the fluorescent ability of AIE active makes it more possible for the drug delivery process and helps to locate the micelles. However, the hydrazone bond cleavage takes place in an acidic environment releasing DOX at the targeted site [137]. They have developed a self-assembled pH-responsive nanoparticle combined with the AIE-active phenomenon. The drugs can be delivered and released to a targeted site at physiological pH which makes DDS be more modified for cytotoxicity with excellent biocompatibility and cell imaging application. A smart pH-responsive with tadpole-shaped PEG POSS-(TPE)_7_ polymer was fabricated by further self-assembling into vesicles and micelles which consists of hydrophobic TPE fluorophore connected by Schiff base bonds. The molecule maintains structural integrity at normal physiological pH but under the acidic condition the aggregated molecules lose their identity. It is noteworthy that caged-shaped POSS improves the AIE effect by restricting the intramolecular rotation of TPE. The polymeric micelle is further encapsulated with antitumor DOX which maintains stability and protects leakage in blood circulation, however, at the same time the rapid release of DOX takes place due to protonation of Schiff base as it gets internalized at the tumor site on the process of drug release mechanism [138]. Hao and co-workers synthesized an amphiphilic conjugated polymeric micelle with 1H-pyrrole-1-propionic acid (MAL)-poly(ethylene glycol) (PEG)-Tripp-bearing comprising of AIE imaging and Forster resonance energy transfer (FRET) which self assembles itself to micelle further loaded with DOX drug. The synthesized polymeric micelle has a 106 nm mean size with excellent stability and high drug loading capacity (10.4%) and 86% encapsulation efficiency. The polymeric micelle consists of Tripp-COOH hydrophobic and MAL-PEG-Tripp amphiphile becomes trapped in micelle core during micellar formation. Initially, MAL-PEG-Tripp showed weak fluorescence but as the fraction of water increases to 99% volume, fluorescent intensity increases 52-fold which confirms further use in a cell imaging application. However, the DOX-loaded MAL-PEG-Tripp micelle showed the FRET phenomenon which was used for tracking the drug release process of DOX-loaded micelle at different pH. It was observed that about 28% of DOX-loaded MAL-PEG-Tripp micelle releases the drug at pH 7.4 and around 64% at pH 5.5 in 24 h of time. The rate of release of the drug was higher at pH 5.5 (90%) compared to that of pH 7.4 (35%) which signifies that the drug release of the DOX-loaded MAL-PEG-Tripp micelle releases the drug at a faster rate at a low pH microenvironment. To study the drug release change, a fluorescence study was carried out which further confirmed the rapid decline in emission wavelength of 500–700 nm for acceptor DOX with a gradual increase in emission wavelength at 470 nm of donor system MAL-PEG-Tripp. This indicated that the FRET process between the DOX and micelle is weakened with the release of the drug from the drug-loaded micelle thus making the FRET process possible to use for monitoring the drug release along with the AIE-active phenomenon [139]. Nanodiamond is an ideal platform for drug delivery because of its unique high surface areas, chemical inert, biocompatibility, and non-toxicity which is functionalized with therapeutics, proteins, antibodies, DNA, and polymers. Herein, Liu et al. developed a fluorescent nanodiamond for bioimaging and drug delivery. The synthesized nano-diamond was non-fluorescent which was further converted to fluorescent nanodiamond by surface medication with AIE active fluorophore with fluorescent polymer via the Diels-Alder reaction (D-A). The fluorescent nanodiamond (ND-poly(Phe-PEGMA-IA)) shows good water fluorescence properties with high water diversity and biocompatibility. However, the nanodiamond can be used as an excellent fluorescent marker. Thereafter the molecule was loaded with an anticancer drug (cisplatin, DDP) wherein the controlled release of the drug mechanism can be well studied under acidic conditions [140]. As discussed earlier, polymeric micelles are in great demand due to their multifunctional property especially for biomedical applications. Su and co-workers developed dual responsive prodrug micelle doxorubicin conjugated amphiphilic PMPC-PAEMA-P (TPE-*co*-HD)-ss-P(TPE-*co*-HD)-PAEMA-PMPC copolymer exhibiting an AIE-active imaging property and charge conversion for chemotherapy and bioimaging application. The imine conjugated with DOX becomes cleaved under an acidic condition which is combined with a hydrophobic core along with the glutathione (GSH) disulfide bond as shown in Figure 26. PAEMA acts as a “gate” that opens in acidic conditions and helps in the drug release mechanism. The behavior of the drug release mechanism was investigated for the DOX conjugated prodrug micelle at varying pH with different concentrations of GSH. It was noted that the micelle showed good stability and at pH 7.4 about 35% of the drug was released from the micelle after 48 h. the conjugated DOX cleaves the imine linkage at pH 6.0 as the PMAE transforms to hydrophilic status, thus at this pH after 48 h it releases 80% of the drug. Thereafter due to pH sensitivity and intracellular redox environment leads to the destruction of the disulfide bond further accelerating drug release by converting the micellar structure; however, 90% of the drug is released at pH 6.0 and in the presence of 10 mM GSH for 12 h. Thus, this makes it a good candidate for dual responsive pH and drug release [141].

Wang et al. reported two color-tunable AIE active conjugated polymers which were synthesized by Pd-catalysis by the Suzuki coupling polymerization reactions P-1 and P-2, with P-1 exhibiting intramolecular FRET pair with clear green colored AIE fluorescence and P-2 with TPE and DTBT moiety turning from green to red fluorescence. This AIE-active polymer can form stable conjugated polymer nanoparticle which acts as drug carrier of paclitaxel (PTX) and showed good cytotoxicity on HeLa and A549 cells which internalize in the cells located with green or red AIE fluorescence. Therefore, this type of AIE-active CPN acts as a drug carrier of PTX which has a self-indicating property with intramolecular FRET pairs [142]. Herein, Gao and co-workers constructed the new system with AIE active carboxylate TPE, benzyl boronic ester (BBE), and prodrug DOX (ABD-system). The ABD system was slightly emissive in solution state as the FRET mechanism from TPE to DOX was more prominent. Upon internalization of the ABD system in the cells the cleavage of the BBE linker takes place along with dissociation of DOX and the TPE unit which further disrupts the FRET mechanism, thus, making it possible for the nucleus to take up the released drug with red fluorescence. Thereafter, the TPE aggregate in cellular plasma exhibited blue fluorescence. The fluorescence changes occurring in blue and red channels have been well investigated, indicating the release of DOX, Figure 27 [143].

Besides the pH-responsive systems, Shamsipur and coworkers designed dual emissive fluorescent silver nanoclusters (AgNCs) capped with hemoglobin exhibiting features such as NC oxidation and AIE enhancement used for drug delivery and cell imaging. The oxidation and AIE-active nature resulted in ligand to metal charge transfer of the NC in presence of oxygen, sulfur, and nitrogen that converts Ag(I) ions to Ag(0)@Ag(I)-Hb core-shell NCs. However, the hyaluronic acid (HA) on the surface of the Hb forms an excellent platform for doxorubicin drugs such as DOX/HA/AgNCs. This system can be used in imaging, gene delivery, biosensing, photocatalysis, and electrochemical applications [144]. Metal-organic AIE active vesicles were developed by using triphenylamine carboxylate as TPA-1@Zn^2+^ which is considered a multifunctional candidate for cell imaging, drug loading, and delivery as well as acting as a “turn-on” sensor for detection of Zn^2+^ ion [46]. Nanomaterial including nanomicelle has potential application in the field of drug delivery. Qian et.al reported a smart nano delivery system (STD) nanomicelle functionalized with peptide and AIE active moiety. The synthesized nanocarrier was modified by using ST which was pH triggered targeting peptide using SKDEEWHKNNFPLSPG sequence and capase-3 modified with DEVD peptide linker comprising AIE-active TPE moiety. However, the TD is the peptide which is a tumor-activated cell-penetrating peptide consisting of TAT (YGRKKRRDRRC sequence) and 2,3-dimethyl maleic anhydride (DA) which further keeps the nanocarrier “Stealth and Stable”. The following STD-NM was further utilized for monitoring drug release at the tumor-targeted site [116]. In addition, DOX-loaded AIE-active nanoparticles were synthesized by Wang and group in 2019 consisting of AIE-active polymeric (FTP) nano material via FRET process that has self-indicating capacity for cancer therapy and utilized for a drug delivery system. In particular, the FTP acts as donor and DOX acts as acceptor which helps in the investigation of a drug release mechanism in acidic conditions during the drug delivery process. In the following work, fluorene (FLU) and TPE were polymerized to give (FLU-TPE). Furthermore, the polyethylene glycol (PEG) was introduced into FLU-TPE to form FLU-TPE-PEG (FTP)**.** The structure of the polymeric material is shown in Figure 28 [145].

Another light activable and AIE-active polymeric nanoparticle for investigation of drug release mechanism. Wu and coworkers employed Pt (IV) prodrug with AIE active fluorophore which was embedded in PtAIECP and DOX further encapsulated with nanoparticle to give PtAIECP@DOX NP. This nanocarrier activated the prodrug Pt (IV) to Pt(II) thus releasing the DOX into the cell via the fluorescence “turn-on” approach resulting in the cleavage of the polymeric linkage. Thus, considered a good platform for Pt (II) as an anticancer and DOX for intracellular drug release at a particular site. This polyprodrug acts as a dual drug release mechanism in combination with AIE active phenomenon. In the reported work, initially upon irradiation a reduction activation process takes place in which Pt(IV) is converted to Pt(II) which further promotes the Pt(II) and DOX to be dissociated. In 1 hour of pre-irradiation, 30% of the Pt and DOX were released making it impossible for the nanocarrier to release the drug in dark conditions. Accordingly, it represents the switch ON/OFF effect in the presence of light and dark conditions. Therefore, the drug release can be investigated by using fluorescent nanoparticles [146].

Dong et al. constructed AIE active self-assembled nanostructure pH/redox with FRET effect that can be utilized for monitoring the drug delivery and release at a specific site. The molecule was synthesized by polyamide amine (H-PAMAM) and polyethylene glycol (PEG) bridged at the periphery by dipropionic acid. Furthermore, this AIE active molecule was loaded with the anticancer drug DOX. It was observed that H-PAMAM shows strong fluorescence, while the AIE effect and FRET between H-PAMAM and DOX help in monitoring the drug release mechanism. As the release of the drug takes place, the FRET process between the carrier and loaded drug completely disappears. Thereafter, the host–guest interaction between the PEG and cyclodextrin (α-CD) forms the nanocarrier as H-PAMAM-ss-mPEG/α-CD abbreviated as (HG/CD) effectively shows both AIE as well as the FRET process. The rapid and selective release of the drug was well monitored by using the pH/redox dual-responsive material carrying excellent biocompatibility enabled the drug carrier tracing, drug release, and chemotherapy [147].

The design and synthesis of multicolored drug carriers have shown significant importance which helps in a self-illuminating drug delivery system. Recently in 2019 Wang and coworkers developed donor-acceptor conjugated polymer exhibiting AIE activity. In this work, the conjugated polymers P1, P2, and P3 were synthesized by using AIE active tetraphenylethylene moiety and donor–acceptor moiety (D-A). The fluorescence characteristics of all three P1, P2, and P3 can be changed from ACQ to AIE by varying the donor moiety Figure 29. The Pdots were loaded with PTX drug prepared by precipitation method. The synthesized Pdots show good stability in the physiological environment. Pdots have high efficacy, biocompatibility and tumor-targeting capacity [148].

Li and group designed multifunctional AIE active 4-*N*,*N*-dimethylaminoaniline salicylaldehyde Schiff-base with different substituents for an intracellular fluorescence assay, imaging, and drug delivery system. The functionalized DSS-4-DEA exhibits high cell-penetrating capacity, which was found to be a more promising fluorescent material. In addition, the DSS-4-DEA fluorescent probe acts as a transmembrane carrier for enhancing the internalization of DSS-5-Cl. Furthermore, the DSS-4-DEA is considered an excellent platform for drug delivery. Here, the fluorescent material is loaded with the anticancer drug curcumin (Cur). The Cur has excellent medical effects including anti-inflammatory and antiproliferative. Since DSS-4-DEA forms the aggregate it can thus be loaded with Cur. Therefore, the DSS-4-DEA displayed multiple bio applications [149]. Ma and group in the year 2019 designed a two-photon fluorescent molecule using DOX and conjugated polymer P(TPMA-co-AEMA)-PEI(DA)-Blink-PEG(PAEE_Blink_-DA) with an AIE-active property was synthesized. This fluorescent molecule has potential application in bioimaging and can be used for theranostic applications. The PEI contains the polyethyleneimine linkage, dimethyl maleic anhydride (DA), and polyethylene glycol (PEG) which is a bridge to the PEI chain via benzoyl imine linkage. This polymeric micelle was further loaded with DOX exhibiting a prolonged-release drug mechanism with bioimaging, and hence considered as an excellent candidate for enhanced chemotherapeutics delivery [150].

There are several traditional multifunctional drug delivery systems (DDS) with a fluorescent dye and targeted unit that were synthesized but due to instability and manufacturing problems as well as low drug is entrapped. Therefore, in 2020 Ma and coworkers constructed multifunctional DDSs via amphiphilic conjugated β-D-galactose with tetraphenylethylene (TPE-Gal) micromolecules. Here, TPE acts as a hydrophobic chromophore while the Gal species acts as a targeting ligand which self assembles and helps in loading water-insoluble paclitaxel (PTX) as well water-soluble DOX anticancer drugs. Thus pH/b-D-galactosidase DOX loaded TPE-Gal@DOX has good antitumor efficacy compared to free DOX [151]. Additionally, another multifunctional AIE active 10-phenyphenothizine (Ph-PTZ) organic fluorescent dye was constructed by using the mesoporous silica nanoparticle with polymeric composite by Huang and co-workers and utilized for the drug delivery system [152]. Gao et al., constructed donor-acceptor-donor (D-A-D) multicationic AIEgens methylpyridium unit’s tetra-pyridium-anchored TPCI, bis-pyridium anchored BPCI, and tetra-ammonium anchored TPCB for unimolecular theranostics application [153]. Zhao and coworkers synthesized AIE active functionalized polymeric gel with a diselenide crosslinker. This diselenide crosslinker becomes fragmented due to H_2_O_2_ or dithiothreitol (DTT). However, the following polymeric gel was encapsulated with DOX and can be used as a good drug carrier with controlled drug delivery and sensing application. Since the redox process is faster in tumor cells, the drug release is explained well under redox conditions as the polymeric gel is well utilized as diselenide bond cleavage takes place for redox stimulus in the tumor environment. Hence, the SeSe*_y_*-PAA-TPE*_x_* can be used for loading and release [154]. Recently, Li and group constructed and fabricated nanoparticles using a self-assembled strategy. The nanoparticles show excellent dual bioimaging function by fluorescence imaging and magnetic resonance imaging (MRI) which further extended their application in targeted cancer therapy. Here, initially the fluorescent vesicle was coated with MRI active Gd^3+^ agent which was considered to be highly active as an anticancer agent. Thereafter, the transferrin (Tf) was coated outside the Gd^3+^ which made the nanoparticle nontoxic for cells. However, charge reversion in lysosomes Tf protein being released makes the nanoparticle toxic and helps in killing the cancerous cells. Thus, this is considered an excellent candidature for theranostic application, Figure 30 [155].

There are several AIE/AIEE possessing luminophore which has extended its application in biomedical applications. So herein Kumar and co-workers have demonstrated different TPE derivative which has wide application in bioimaging and utilized further for drug delivery process [156]. Recently Yan et.al designed a drug carrier based on hyaluronic acid-tetraphenyl ethylene conjugate (HA-SS-TPE) comprising of glutathione (GSH) for drug delivery. The polymeric micelle was further loaded with DOX, however, the glutathione helps to release this DOX at the targeted site. The DOX-loaded DOX@HA-SS-TPE completely suppressed tumor-related growth. The cellular uptake and drug delivery was very well investigated for the DOX@HA-SS-TPE micelle. Red fluorescence was observed as the DOX is released at the nucleus. Hence the results demonstrate that the polymeric micelle is highly effective and helps in the drug release mechanism [157]. Sun and group designed polymeric multifunctional AIE active redox-sensitive nanocarrier for bioimaging which can be used for a drug delivery system. The polymeric micelle was synthesized by using AIE fluorophore Tripp to methoxy-PEG with a redox-sensitive disulfide bond. Thereafter, the polymer self assembles to a micelle wherein mPEG proves to be a biocompatible shell for long-term circulation of blood, and AIE-active fluorophore acts as a fluorescent marker for cellular imaging and finally loading of hydrophobic DOX. Thus, making the disulfide bond play a major role in the release of the drug at the targeted tumor site. The drug-loaded micelle was investigated on 4T1 cells by CCK-8 [158]. Here, another TPE functionalized polymeric micelle modified chitosan (TPE-bi(SS-CS-Bio) which has good potential for cellular imaging, targeted drug delivery, and controlled release of paclitaxel. The polymeric micelle has a high drug loading capacity and self assembles into a micelle. The polymeric micelle is loaded with paclitaxel drug in vitro and a study revealed that the glutathione micelle disassembles itself. Moreover, the polymeric drug-loaded micelle exhibited excellent cytotoxicity towards the MCF-7 cells. The following polymeric PTX loaded micelle was investigated for antitumor activity on mice to inhibit tumor growth [159]. Gu and co-workers designed TPE bases AIE-active soft dots through the self-assembly approach. The dots TPA-TPE-OEG-N_3_ were synthesized by using fluorescent core and oligo ethylene glycol (OEG) chain with azide group which self-assemble into a nanoparticle. Moreover, these soft AIE active dots were loaded with unmethylated cytosine-phosphate-guanine (CpG) that induces the immunostimulatory effect of RAW264.7 cells. Here, TPA-TPE acts as a hydrophobic core, wherein the OEG azide acts as a hydrophilic substitute, and -N3 via a click reaction reacts to the drug. Therefore, the synthesized probe is an excellent platform for the drug delivery system [160]. Interestingly, Liu et al. constructed an AIE-active pillar[5]arene [H] based host-guest nanoparticle having an excellent advantage in bioimaging and drug delivery process [161].

## 5. Conclusions and Perspectives

The recent developments in the fluorescent materials in solid and aggregated forms have shown importance in the different scientific and technological fields, especially in biomolecular and environmental applications. There are several organic fluorescent molecules but they suffer from the detrimental phenomenon known as aggregation-caused quenching. To overcome the ACQ process, the Tang group developed another phenomenon known as AIE which is now commonly and widely used in many fields. The AIE phenomenon has become an excellent platform for sensing, biological cell imaging, and drug delivery systems. TPE is the most frequently used luminogen upon functionalization for various purposes to its AIE phenomenon and mechanochromic properties. Nevertheless, AIEgens have been widely used in green energy devices and environmental monitoring. Furthermore, the application can be explored in food safety and quality control which is a major focus in terms of public health.

Despite remarkable advancement and development in the AIEgens in sensing, cell imaging, and drug delivery, some challenges should be addressed to find solutions to remaining hurdles. There are a few disadvantages attributed to them such as multiple-step synthesis, non-specific features due to binding sites, and non-targeted drug delivery.

To overcome these small issues, the rational design of AIEgens should comprise specific binding sites, which can be used as receptor/binding sites for high selectivity, sensitivity towards the particular analyte, along with easy and simple synthetic routes for design the AIEgens. One more aspect to keep in mind is that the photostability and solubility for a specific application, thus, there is a need for simple and cost-effective strategies to be employed. Therefore, given these facts fluorescent materials need more attention in terms of good stability with varying functionalities.

In conclusion, we have focused on the most recent AIEgens and their potential application in the field of sensing, cell imaging, and biomedical application, especially in drug delivery systems. Although several luminogens have been successfully employed limits to their application remain and they are yet to be exploited. In this review, we have described briefly the different AIE-active molecules especially TPE, pyrene, salicylaldehyde organic dye, metal-organic framework, polymeric micelles for sensing of essential, non-/toxic metal ions, and anions. This review also describes AIE luminogens for cell imaging and drug delivery systems. Selectivity and specificity are most important factors considered in biomedical application which can be achieved by combining the target-specific moiety to AIE active chromophore. Thus, distinct properties of AIEgens make them available for sensing, cell imaging and theranostic application. However, more insightful research in the field of AIE-active materials is required for further application in the field of supramolecular chemistry.

Even though AIEgens has extended its application in cell imaging and drug delivery systems, there are many problems still to overcome, for example, short wavelengths absorption, non-optimized emission, and broad emission spectrum which restricts its usage in biomedical applications. To overcome the aforementioned problems, the multiphoton absorption, harmonic generation, and up-conversion methods should be constantly employed.

## Figures and Tables

**Figure 1 molecules-27-00150-f001:**
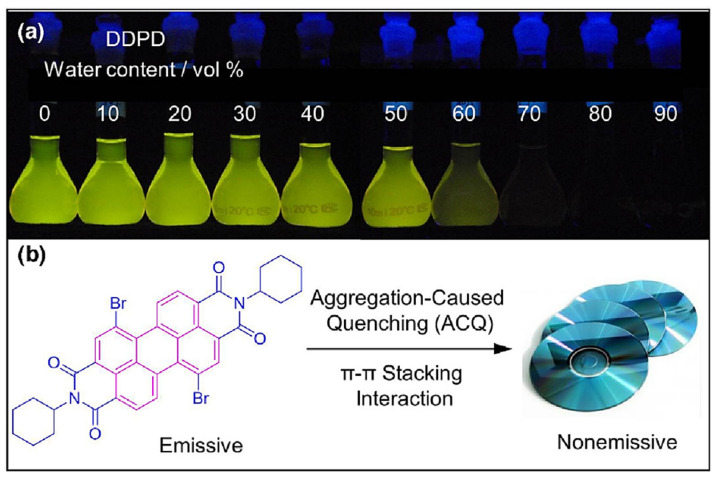
(**a**) Fluorescence photograph of the (*N,N*-dicyclohexyl-1,7-dibromo-3,4,9,10-perylenetetracarboxylic diimide) molecule with increasing water content and (**b**) showing the aggregation-caused quenching (ACQ) phenomenon. Reprinted from reference [13] with the permission of the Royal Chemical Society.

**Figure 2 molecules-27-00150-f002:**
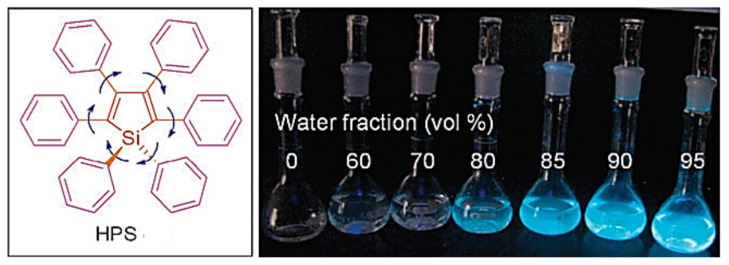
Photograph of hexaphenylsilole (HPS) representing aggregation induced emission (AIE) phenomenon. Reprinted from reference [12] with the permission of the Royal Society of Chemistry.

**Figure 3 molecules-27-00150-f003:**
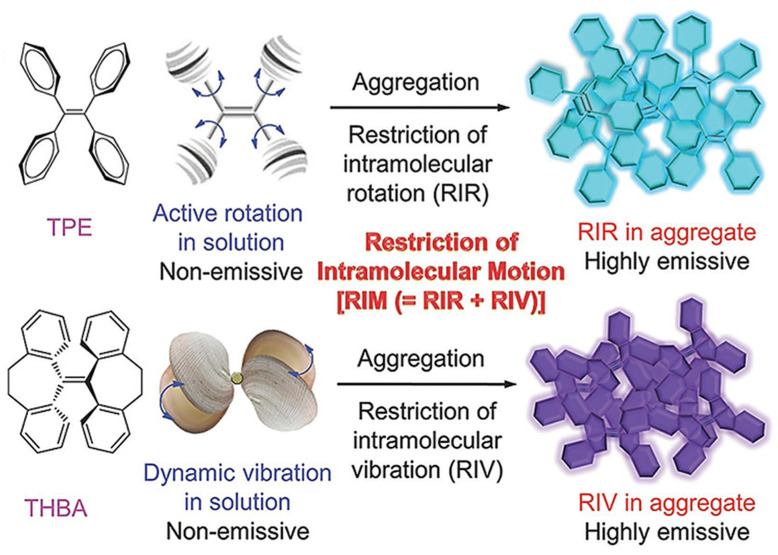
Representation of proposed mechanism for AIE effect. Reprinted from reference [29] with the permission of Wiley-VCH.

**Figure 4 molecules-27-00150-f004:**
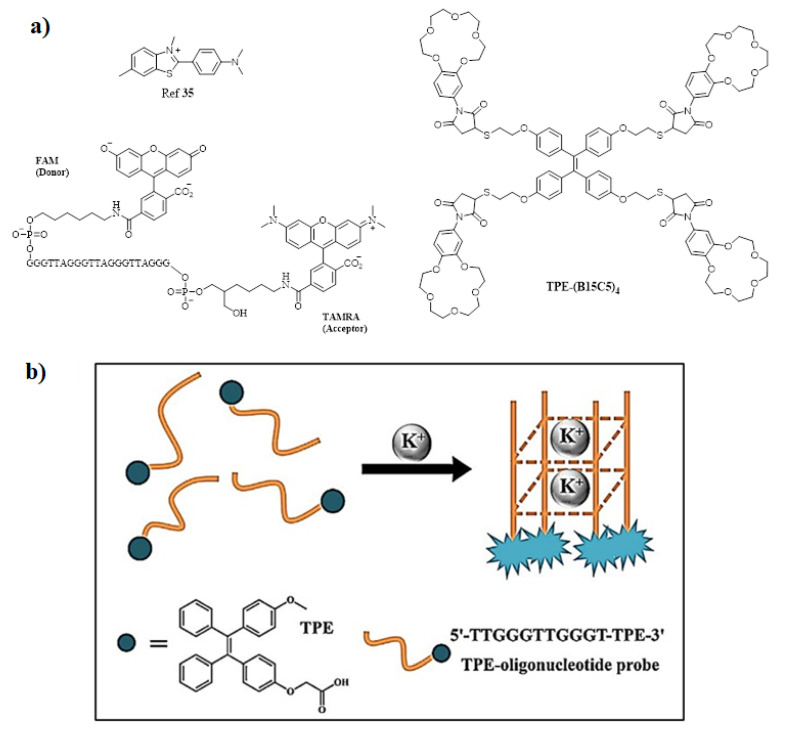
(**a**) Structural illustration of the synthesized AIE active fluorescent molecules for sensing of K^+^ [38] (**b**) tetraphenylethyne (TPE) modified DNA oligonucleotide used for sensing of K^+^. Reprinted from reference [36] with the permission of Elsevier.

**Figure 5 molecules-27-00150-f005:**
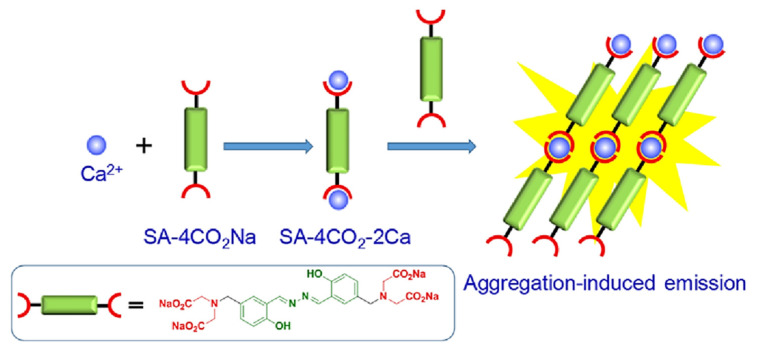
Schematic illustration of the AIE active probe SA-4CO_2_Na for intracellular detection of Ca^2+^**.** Reprinted from reference [40] with the permission of the American Chemical Society.

**Figure 6 molecules-27-00150-f006:**
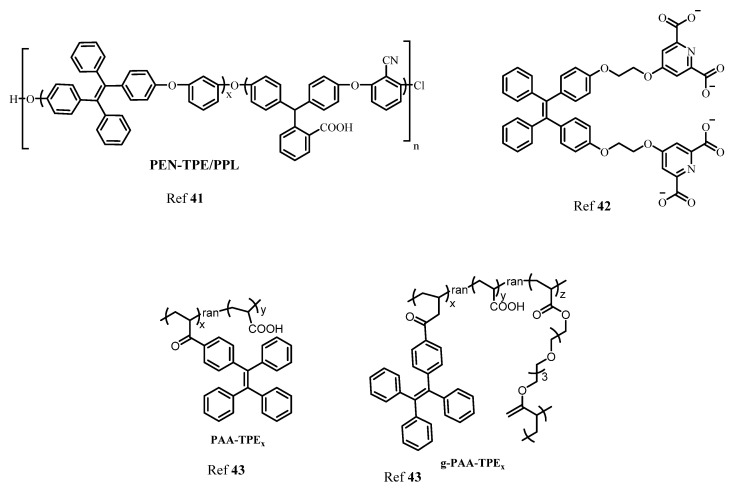
The structures of different AIE active TPE derivatives for sensing of Ca^2+^.

**Figure 7 molecules-27-00150-f007:**
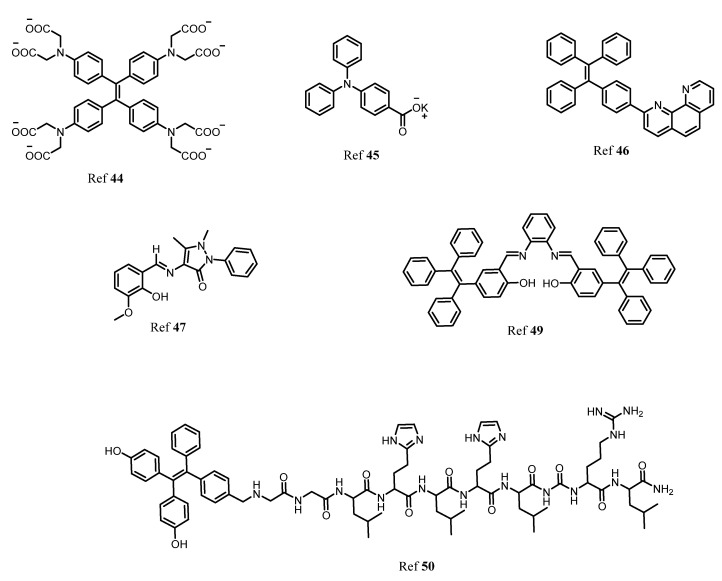
The various AIE active fluorophores illustrated for detection of Zn^2+^ ion in solution.

**Figure 8 molecules-27-00150-f008:**
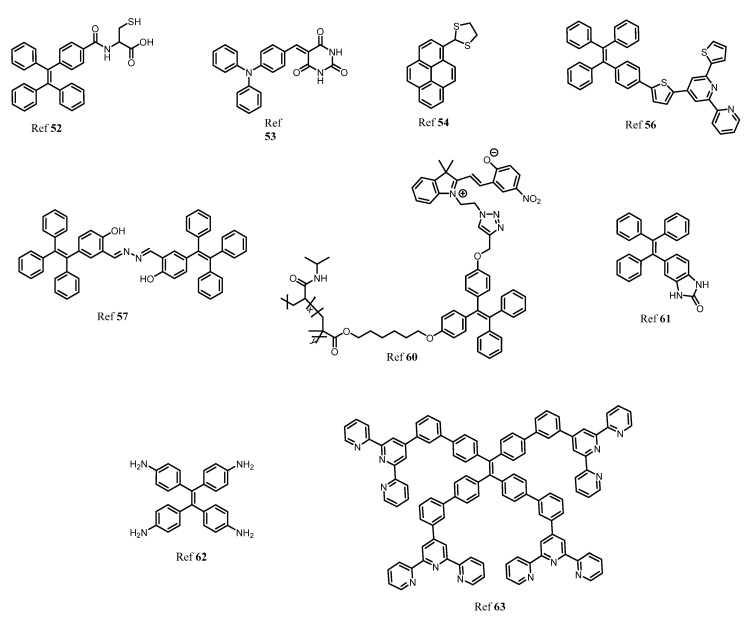
The structural representation and examples of several substituted AIE active fluorescent materials for sensing non-essential metal ions and anions.

**Figure 9 molecules-27-00150-f009:**
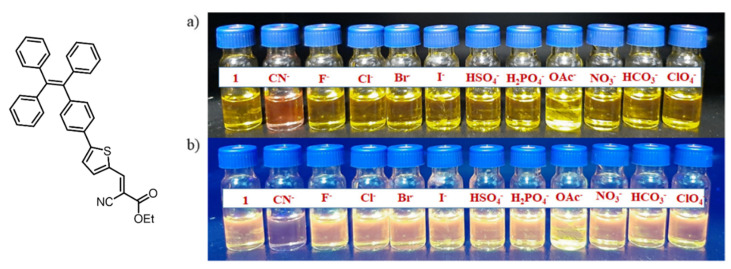
A photograph representing the sensing performance of the probe towards CN^−^ ion in an aqueous solution in the presence of other anions under: (**a**) visible light and (**b**) UV light (365 nm). Reprinted from reference [60] with the permission of the American Chemical Society.

**Figure 10 molecules-27-00150-f010:**
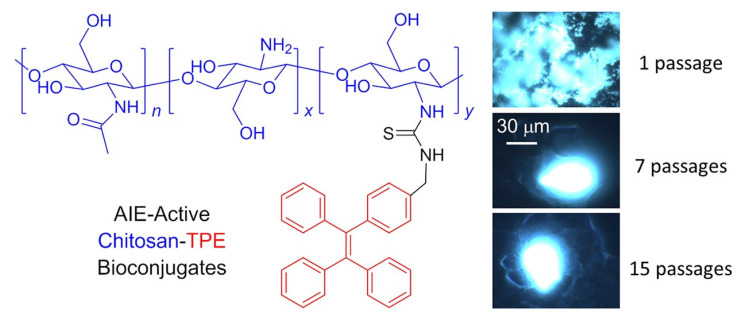
Structural illustration and utilization of TPE-CS as an excellent fluorescent marker in biological cell imaging in HeLa cells. Reprinted from reference [76] with the permission of the Journal of the American Chemical Society.

**Figure 11 molecules-27-00150-f011:**
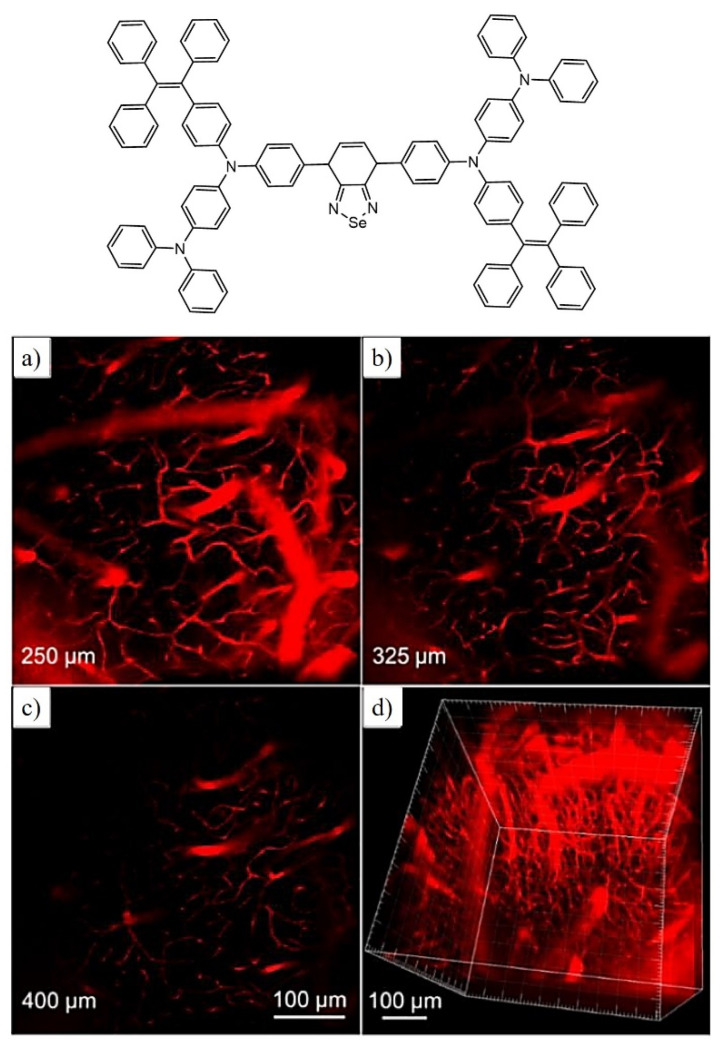
Representative molecular structure with fluorescent images captured under two-photon imaging technique for brain blood vessels upon incubation of mouse cells for 0.5 h with TTSe dots (**a**–**c**). (**d**) represents a high-resolution 3-D image of blood vessels at 100 μM scale. Reprinted from reference [81] with the permission of The Royal Society of Chemistry.

**Figure 12 molecules-27-00150-f012:**
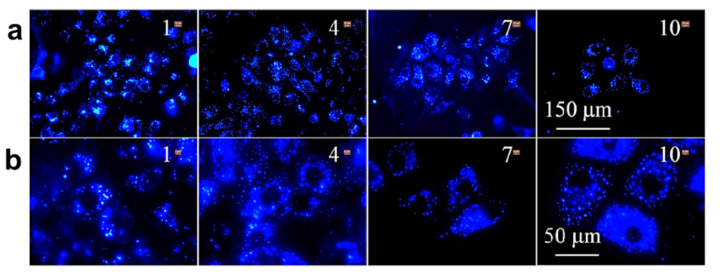
The fluorescence images of A549 cells upon incubation with a P6 probe (100 μg/mL) were captured at magnification (**a**) ×40 and (**b**) ×100. Reprinted from reference [84] with the permission of the American Chemical Society.

**Figure 13 molecules-27-00150-f013:**
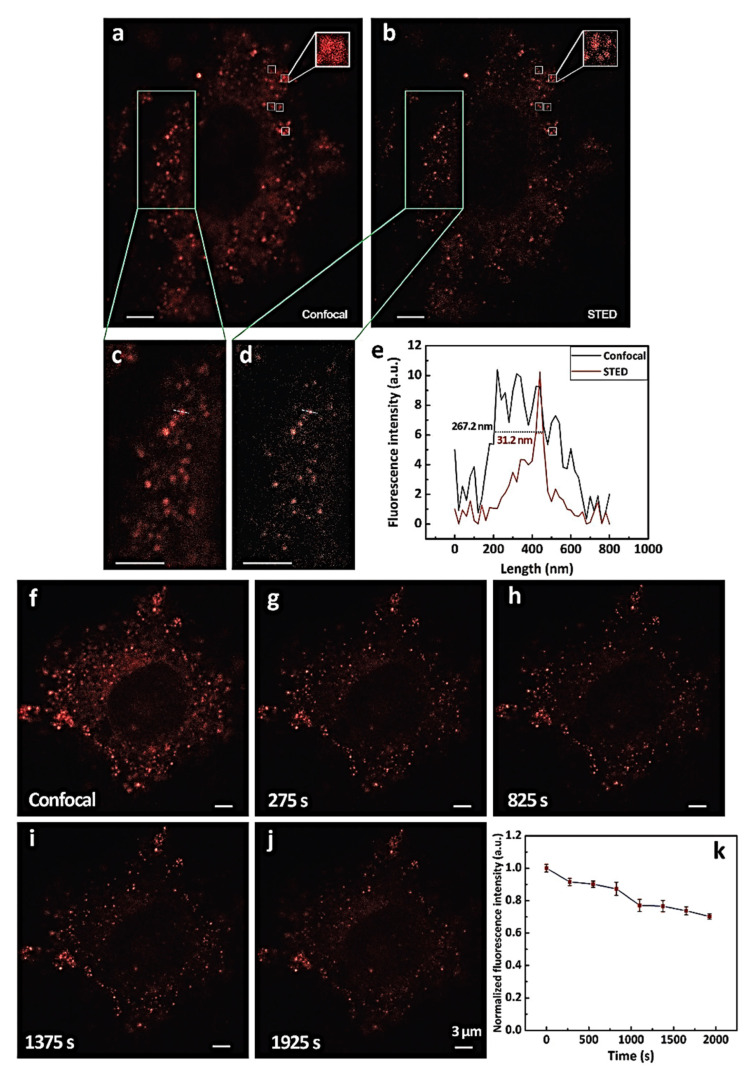
Fluorescence images of HeLa cells stained with TTF@SiO_2_ NPs observed under confocal microscopy and STED nanoscopy imaging. (**a**,**b**) Confocal and STED nanoscopy images of cells incubated with TTF@SiO_2_ NPs while (**c**,**d**) represents the magnified fluorescent image of (**a**,**b**). (**e**) The fluorescence intensity recorded by confocal (black) and STED (red). (**f**). The image demonstrates the confocal microscopy image for TTF@SiO_2_ NPs for HeLa cells and from (**g**–**j**) STED nanoscopy images captured at different time points for HeLa cells. (**k**) The normalized fluorescence intensity for HeLa cells incubated with TTF@Si_2_ NPs at different time points. Reprinted from reference [88] with permission from Wiley-VCH.

**Figure 14 molecules-27-00150-f014:**
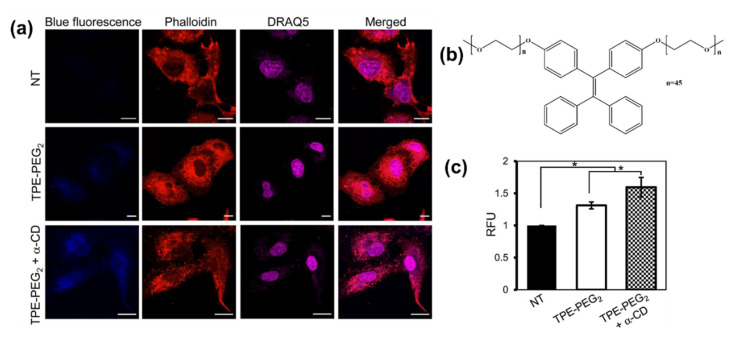
(**a**) Confocal images illustrating the cellular internalization of the TPE-PEG_2_ and combining TPE-PEG_2_ and α-CD using A549 cells and investigated for 4 h at 37 °C. (**b**) Molecular structure of the TPE-PEG_2_ and (**c**) bar graph image representing relative fluorescence intensity TPE-PEG_2_ in respective cells, in which “*” represents three independent experiments from five random field per experiments i.e. * *p* < 0.05. Reprinted from reference [90] with the permission American Chemical Society.

**Figure 15 molecules-27-00150-f015:**
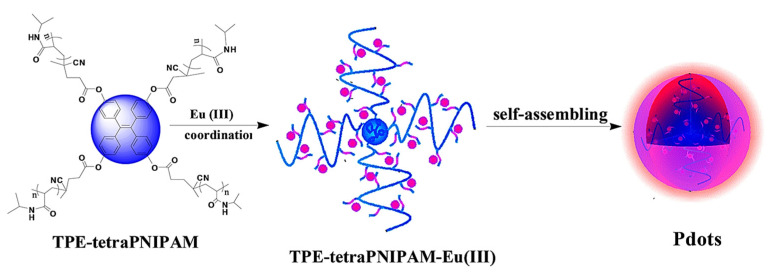
The molecular structure of AIE active Pdots TPE-tetraPNIPAM-Eu (III) used for imaging of cancer cells.

**Figure 16 molecules-27-00150-f016:**
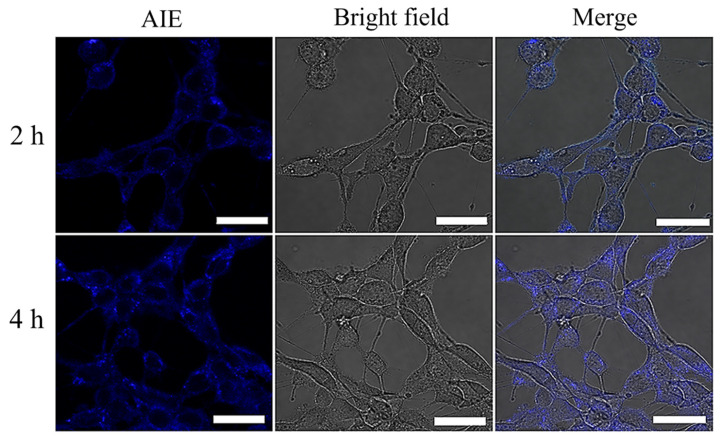
Photograph demonstrating the confocal fluorescence images captured after incubation of 4T1 cells with m-PEG-P(TPE-*co*-AEMA) probe for 2 to 4 h. Reprinted from reference [92] with the permission of the American Chemical Society.

**Figure 17 molecules-27-00150-f017:**
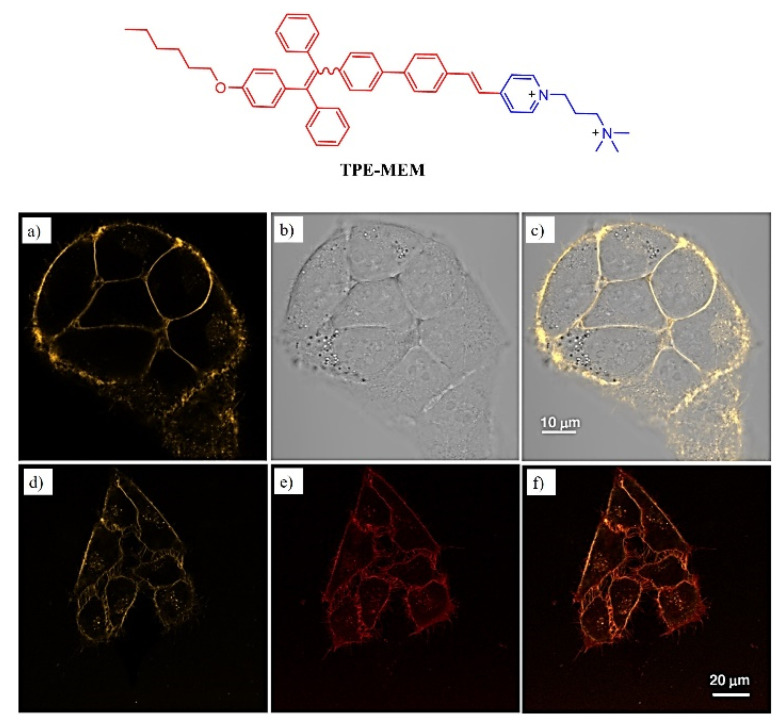
The structure of the fluorescent marker probe (TPE-MEM) imaged under confocal microscopy was captured for TPE-MEM upon incubation of HeLa cells. (**a**) Images under laser scanning confocal microscopy, (**b**) images under bright field, and (**c**) merged images for HeLa cells stained with the TPE-MEM. Image of HeLa cells co-stained with (**d**) TPE-MEM and (**e**) the CellMask Deep Red plasma membrane and followed by image. Further, (**f**) shows merged fluorescence images of ‘a & b’. Reprinted from reference [93] with the permission of American Chemical Society.

**Figure 18 molecules-27-00150-f018:**
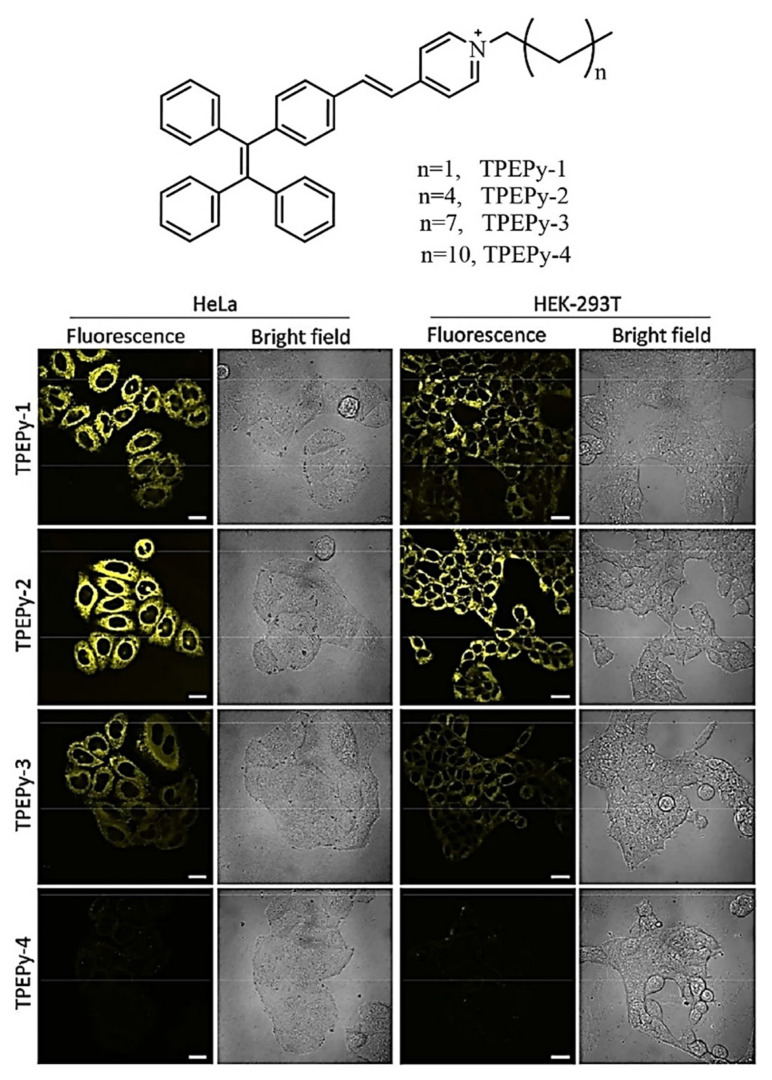
Structural illustration of molecule and CLSM images captured for HeLa cells and HEK-293T cells upon incubation with (TPEPy-1, TPEPy-2, TPEPy-3, TPEPy-4) fluorogens. Reprinted from reference [95] with the permission of the American Chemical Society.

**Figure 19 molecules-27-00150-f019:**
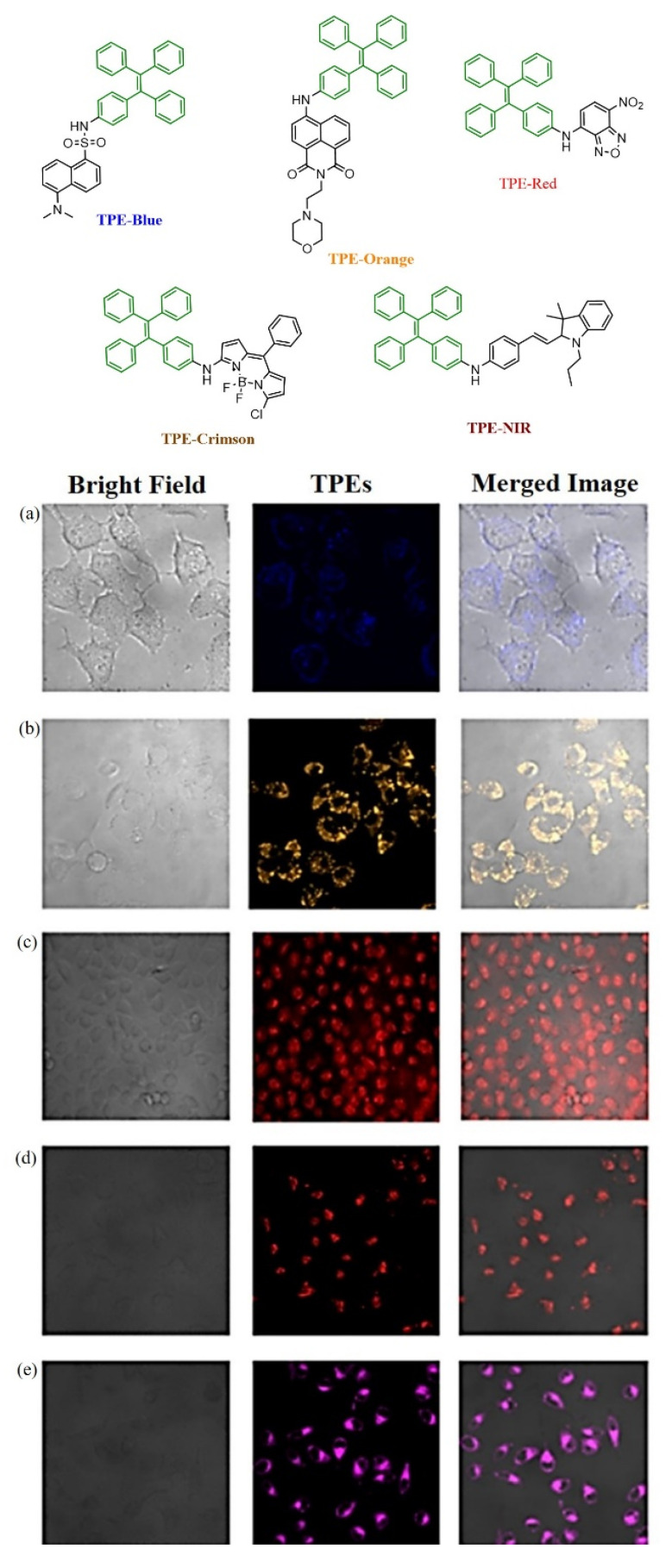
Structures of the fluorescent molecules TPE-NIR, TPE-Blue, TPE-Crimson, TPE-Orange, and TPE-Red are used for biological cell imaging. Confocal images of Hela cells in different emission channels with TPE-Blue (**a**), TPE-Orange (**b**), TPE-Red (**c**), TPE-Crimson (**d**), and TPE-NIR (**e**) (the near-infrared fluorescence was modified graphically to purple for visual clarity). Reprinted from reference [98] with the permission.

**Figure 20 molecules-27-00150-f020:**
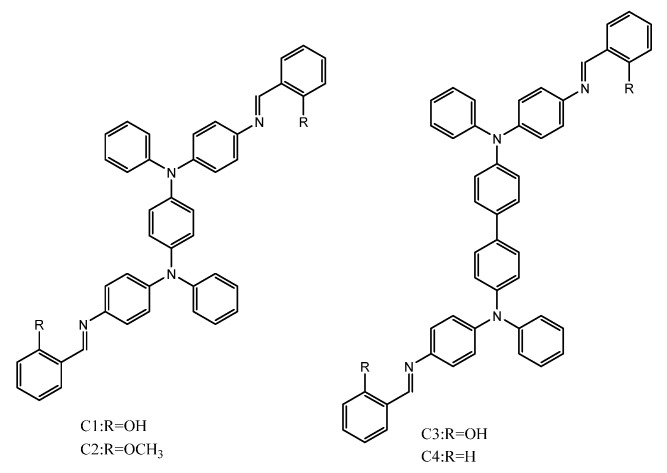
Structural representation of the molecules for detection of hydrazine in live cells.

**Figure 21 molecules-27-00150-f021:**
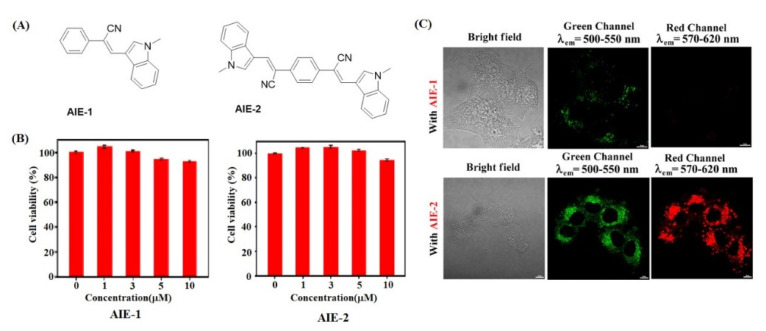
(**A**) Represents the structure of the AIE-1 and AIE-2 probes. (**B**) The plot demonstrates the cytotoxicity assay for HeLa cells incubated with AIE-1 and AIE-2 for 24 h. (**C**) Confocal fluorescence images were captured for HeLa cells incubated with AIE-1 and AIE-2 fluorescent probes. Reprinted from reference [100] with the permission of MDPI.

**Figure 22 molecules-27-00150-f022:**
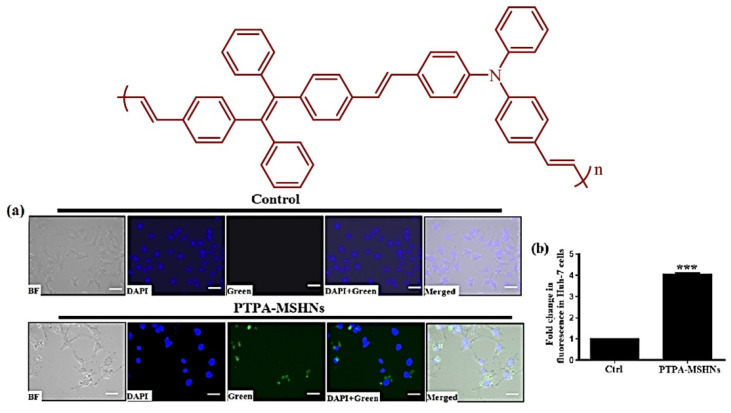
Demonstrates the structure of the fluorescent PTPA luminogenic probe. (**a**) Fluorescent microscopic images of cells when treated with PTPA-MSHNs (100 μg/mL) for 24 h. In cells marked as “control”, shows no addition of PTPA-MSHN; BF stands for bright field; to stain the nucleus DAPI counter stain were used. Scale bar represents 20 μm. (**b**) Quantification of fluorescence emitted by Huh-7 cells after treatment with PTPA-MSHNs (100 μg/mL) for 24 h. “Ctrl” represents cells without treatment of PTPA-MSHN. The symbol (***) represents a significant difference as compared to untreated control. Reprinted from reference [105] with the permission of the American Chemical Society.

**Figure 23 molecules-27-00150-f023:**
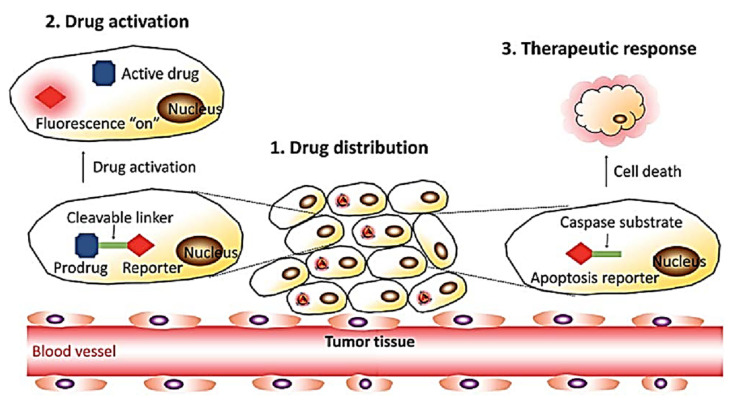
Diagrammatic representation for the AIEgens utilized for therapeutic approach involved in drug distribution and drug activation in situ via drug delivery system (DDS)-based molecules. Reprinted from reference [130] with the permission of The Royal Society of Chemistry.

**Figure 24 molecules-27-00150-f024:**
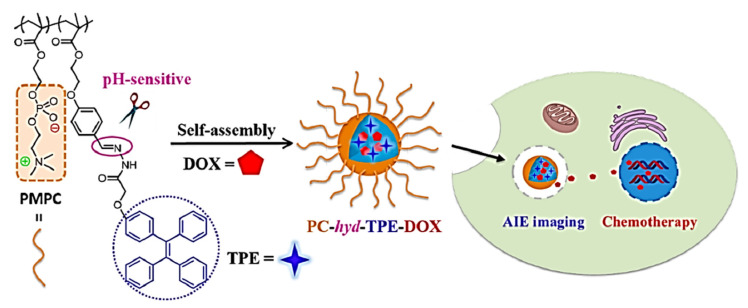
Structure of the molecule PC-*hyd*-TPE-DOX representing application for drug release and cell imaging. Reprinted from reference [135] with the permission of the American Chemical Society.

**Figure 25 molecules-27-00150-f025:**
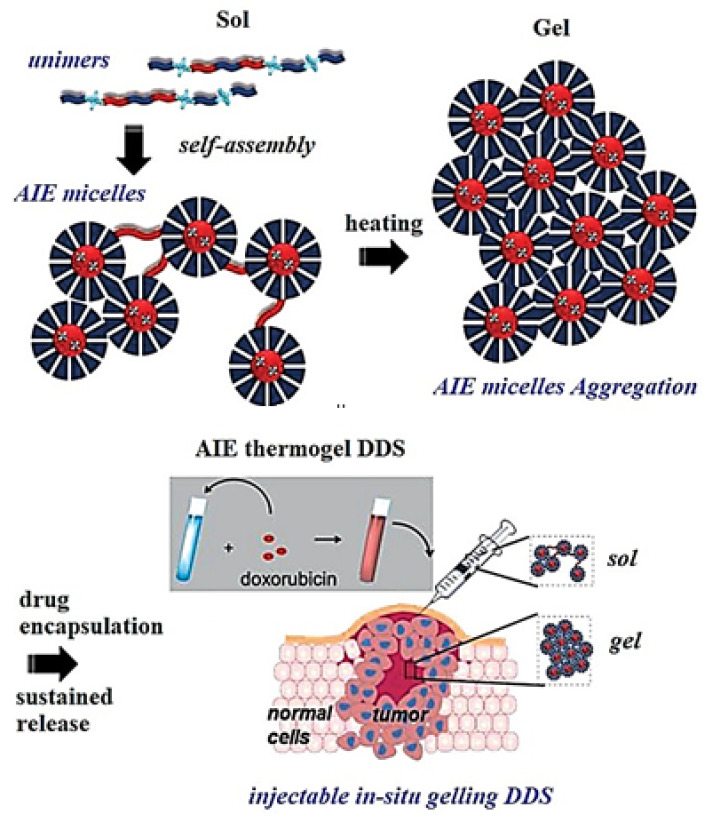
Schematic representation of unimers, AIE-active micelles, aggregation of a micelle, and in-situ gelling of drug delivery system. Reprinted from reference [136] with the permission of Wiley-VCH.

**Figure 26 molecules-27-00150-f026:**
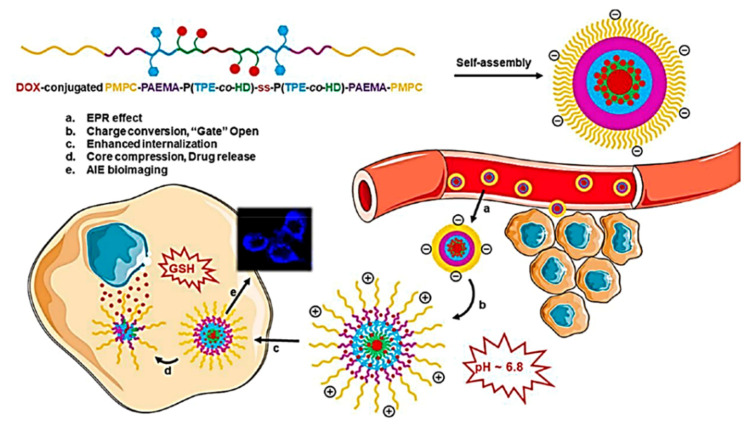
Diagrammatic illustration of polymeric micelle with DOX conjugated PMPC-PAEMA-P(TPE-*co*-HD)-ss-P (TPE-*co*-HD)-PAEMA-PMPC controlled pH and redox responsive release of drug, charge conversion, and bioimaging application. Reprinted from reference [141] with the permission of the American Chemical Society.

**Figure 27 molecules-27-00150-f027:**
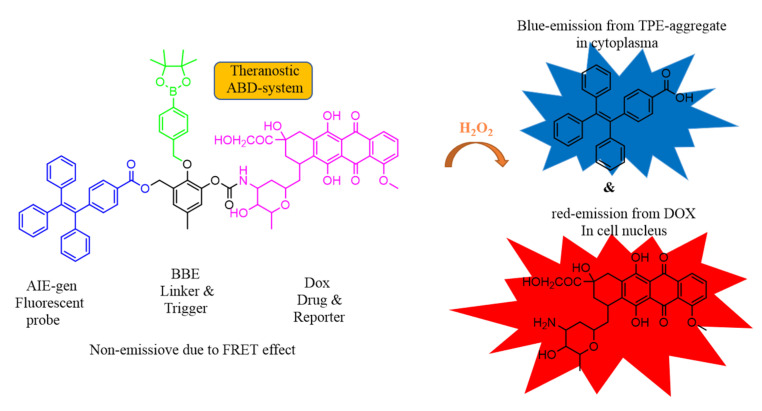
Representative chemical structures of the DDSs system along with working mechanism towards the response to H_2_O_2_. Reprinted from [143] with the permission of The Royal Society of Chemistry.

**Figure 28 molecules-27-00150-f028:**
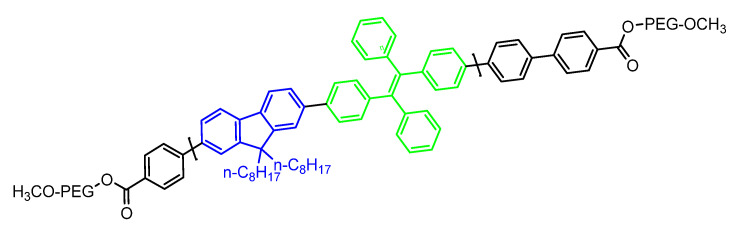
Molecular structure of the compound FLU-TPE-PEG.

**Figure 29 molecules-27-00150-f029:**
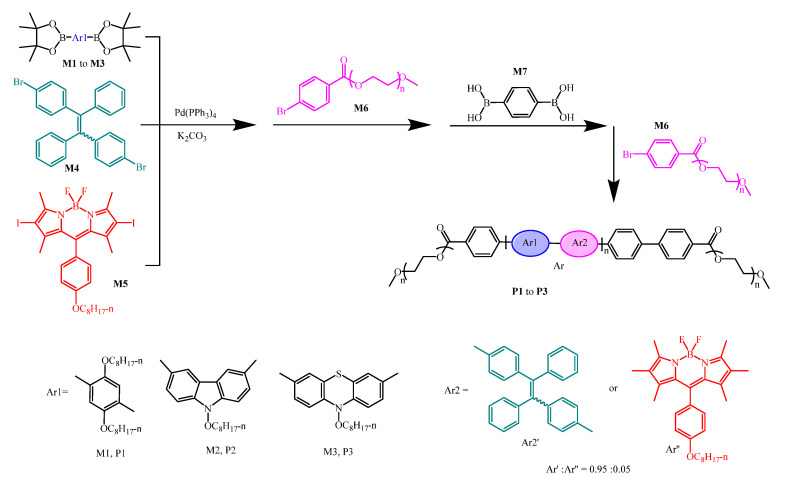
Schematic representation of multicolored AIE-active polymeric dots (Pdots) for drug delivery.

**Figure 30 molecules-27-00150-f030:**
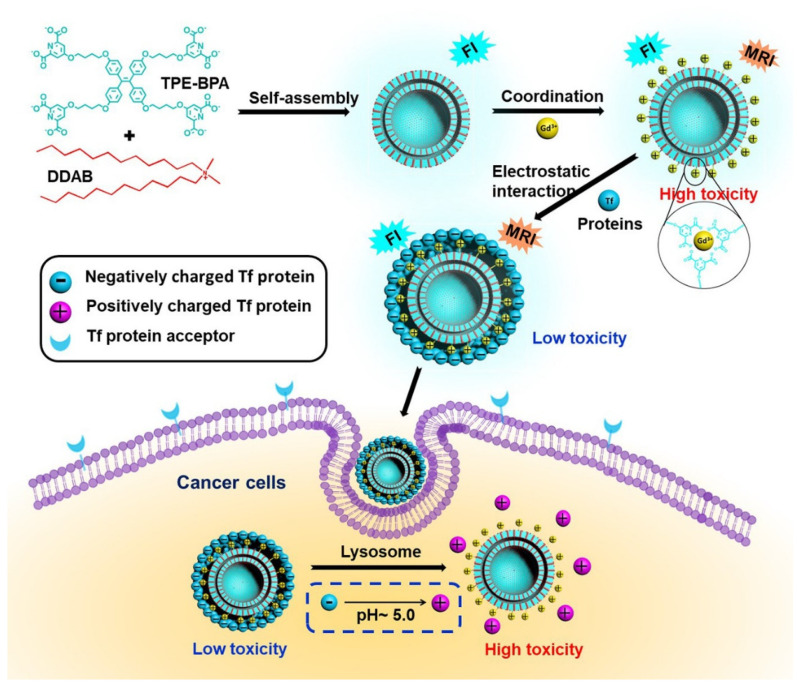
Systematic representation of self-assembled multifunctional AIE active Tf-coated vesicle for fluorescence imaging and targeted cancer therapy. Reprinted from reference [155] with the permission of the American Chemical Society.

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
