# Peer review of "Recent Advances in Aggregation-Induced Emission Active Materials for Sensing of Biologically Important Molecules and Drug Delivery System"

_molecules, 2021, doi:10.3390/molecules27010150_

Round 1
Reviewer 1 Report
Zalmi G. et al have reviewed the recent advances in the development of aggregation induced emission materials and their applications as sensors. Specifically, the sensing application in ions detection, biological cell imaging and drug delivery system has been thoroughly discussed. The reviewer recommends acceptance of the review paper after a major revision.
- In the introduction part, the authors mainly discuss aggregation caused quenching (ACQ) and aggregation induced emission (AIE). However, the relaxation of the excited state is rather complex in solution and in the solid state, including not only the radiative decay, but also the non-radiative decay. The authors should give a more comprehensive discussion for all the possible decay channels, then highlight AIE.
- In the 5 section, the authors mentioned that one disadvantage of AIEgens is the challenge in synthesis. Can the authors also comment the possibility of “rational design” of AIEgens in the current research stage? What kind of designing principles have been proposed to realize the “rational design” for a specific application?
- Many typos should be corrected, the authors should really carefully read their manuscript before submission. Some typos are listed below:
Page 2, “Tang group in 2001 and the group to shows overcome of the ACQ process -so”…
Page 2, “The mechanism involved and well accepted is the restriction of intramolecular rotation, along with the rotation and vibration mechanism that explains very well the motions involved in AIE [21,22].” This sentence is hard to understand.
Page 2, “application, solar cells cell imaging”, solar cells cell?
Page 2, “shows different ““turn-on” ” “turn-on” sensing mechanism”, hard to understand.
Page 2, “Yet another approach via destruction of photoinduced electron (PET), intramolecular charge transfer (ICT) as well as electron transfer”, hard to understand.
Page 2, “In this review, , we” two commas?
Figure 2, captain, “Royal Society Of Chemistry” should be “Royal Society of Chemistry”
Page 4, “Neverthless, small organic molecules having fluorescent properties shows to have advantageous over other reported chromophores due to their high sensitivity and selectivity, high quantum yield, simple synthetic routes, easy operation, and real-time detection is also possible [31].” This sentence is confusing.
Page 7, “Infact” should be “In fact”
Page 9, “Most recently, our group has designed and synthesized TPE based AIE active fluorescent probe which is highly selective and sensitive towards cyanide.It can be clearly seen that the probe can be efficiently utilized for naked-eye detection (Fig. 8),”. There should be a space at the end of the sentence. The way to refer figures should be consistent in the whole text.
Page 12, “The various fundamental processes taking place in life can be very well studied and understood which can be achieved effortlessly using fluorescence tool that has now become one of the most important and launched in biological research for advance and a better understanding of various intracellular processes and dynamics at a cellular level.” This sentence is hart to follow.
Page 13, “In Wan group work” should be “In Wan’s group’s work”, or “In the work from Wan’s group”.
Page 14, “They have shows” should be “They have shown”.
Page 14, “may also can be”, wrong in grammar.
Page 14, “wasfabricated” should be “was fabricated”.
Page 16, “were evaluated and confirms” wrong in grammar.
Page 16, “by Liow and group”, some contents were missed.
Page 18, “at room temperature with 5μM, upon removal of the excess of the probe by washing with PBS solution.” Some contents were missed.
Page 22, “they habe noticed that” should be “they have noticed that”
Page 22, “Wang and group synthesized”, some contents were missed.
Page 22, “hydrazine N2H4.H2O in living cells”, the dot in the formula should be put in the middle.
Page 23, “D-п-A system which id further incorporated into” should be “D-п-A system which is further incorporated into”.
Author Response
Zalmi G. et al have reviewed the recent advances in the development of aggregation induced emission materials and their applications as sensors. Specifically, the sensing application in ions detection, biological cell imaging and drug delivery system has been thoroughly discussed. The reviewer recommends acceptance of the review paper after a major revision.
Response: Thank you for finding our work suitable for publications in Molecules
- In the introduction part, the authors mainly discuss aggregation caused quenching (ACQ) and aggregation induced emission (AIE). However, the relaxation of the excited state is rather complex in solution and in the solid state, including not only the radiative decay, but also the non-radiative decay. The authors should give a more comprehensive discussion for all the possible decay channels, then highlight AIE.
Response: Thanks, we have done the comprehensive discussion of AIE molecules
- In the 5 section, the authors mentioned that one disadvantage of AIEgens is the challenge in synthesis. Can the authors also comment the possibility of “rational design” of AIEgens in the current research stage? What kind of designing principles have been proposed to realize the “rational design” for a specific application?
Response: Thank you, we have incorporated our changes in section-5
- Many typos should be corrected; the authors should really carefully read their manuscript before submission. Some typos are listed below:
Page 2, “Tang group in 2001 and the group to shows overcome of the ACQ process -so”
Page 2, “The mechanism involved and well accepted is the restriction of intramolecular rotation, along with the rotation and vibration mechanism that explains very well the motions involved in AIE [21,22].” This sentence is hard to understand.
Page 2, “application, solar cells cell imaging”, solar cells cell?
Page 2, “shows different ““turn-on” “turn-on” sensing mechanism”, hard to understand.
Page 2, “Yet another approach via destruction of photoinduced electron (PET), intramolecular charge transfer (ICT) as well as electron transfer”, hard to understand.
Page 2, “In this review, we” two commas?
Figure 2, captain, “Royal Society of Chemistry” should be “Royal Society of Chemistry”
Page 4, “Neverthless, small organic molecules having fluorescent properties shows to have advantageous over other reported chromophores due to their high sensitivity and selectivity, high quantum yield, simple synthetic routes, easy operation, and real-time detection is also possible [31].” This sentence is confusing.
Page 7, “Infact” should be “In fact”
Page 9, “Most recently, our group has designed and synthesized TPE based AIE active fluorescent probe which is highly selective and sensitive towards cyanide.It can be clearly seen that the probe can be efficiently utilized for naked-eye detection (Fig. 8),”. There should be a space at the end of the sentence. The way to refer figures should be consistent in the whole text.
Page 12, “The various fundamental processes taking place in life can be very well studied and understood which can be achieved effortlessly using fluorescence tool that has now become one of the most important and launched in biological research for advance and a better understanding of various intracellular processes and dynamics at a cellular level.” This sentence is hart to follow.
Page 13, “In Wan group work” should be “In Wan’s group’s work”, or “In the work from Wan’s group”.
Page 14, “They have shown” should be “They have shown”.
Page 14, “may also can be”, wrong in grammar.
Page 14, “wasfabricated” should be “was fabricated”.
Page 16, “were evaluated and confirms” wrong in grammar.
Page 16, “by Liow and group”, some contents were missed.
Page 18, “at room temperature with 5μM, upon removal of the excess of the probe by washing with PBS solution.” Some contents were missed.
Page 22, “they habe noticed that” should be “they have noticed that”
Page 22, “Wang and group synthesized”, some contents were missed.
Page 22, “hydrazine N2H4.H2O in living cells”, the dot in the formula should be put in the middle.
Page 23, “D-п-A system which id further incorporated into” should be “D-п-A system which is further incorporated into”.
Reviewer 2 Report
This manuscript introduces AIEgens' latest developments in sensing, cell imaging, and Drug Delivery System. Depending on the relationship between the structure and function of AIEgens, the design strategies of AIEgens with different functions are summarized. This review is of significance for possible readers to understand the AIE materials in the field of biological system. However, this manuscript was written badly, and therefore I suggest a major revision before considering a publication of this review. Some concerns are given as follows.
- ‘Sensing’ has been applied in many fields, such as biological system in this review, environment, life, etc. This review focus the sensing in biological system, and therefore, the authors should use “biological sensing” or “biosensing” in the title and subtitle.
- In fact, there are many review papers summarizing AIE system, but what is the difference between this review and other reviews? The authors should introduce the difference in Introduction.
- In page 2, I can understand that the abbreviation ‘RIV’ means ‘restriction of intramolecular vibration’ as marked in the picture, but the abbreviation is not mentioned in the text. Please check it.
- In page 5, does ‘The abnormal level of Ca in the body’ refer to the total calcium in the body or the Ca2+ level in the body, please verify.
- For the reference [40] in the review (from page 5 to page 6), the manuscript introduced the synthesis method of SA-4CO2Na, but the mechanism for SA-4CO2Na detecting Ca2+ was missing. However, the corresponding Figure 5 shows the detection mechanism, so I advise to add the detection mechanism in the text.
- The source of many harmful ions and their detection methods are mentioned in page 8, but no cited reference is mentioned. Please add the corresponding reference to increase the credibility.
- In page 9, copper is an essential trace element for the human body and should not be simply classified as a toxic ion. Therefore, please confirm whether it is reasonable to classify copper ion detection as ‘toxic ion detection’.
- In pages 11 to 12, ‘Using the similar criteria of FR/NIR Qin and the group synthesized selenium-containing FR/NIR AIE active luminogen (TTSe dots) which are very rare however there are no reports available on selenium-containing fluorescent probes for bioimaging application.’ this sentence is vague and easily misunderstood. Moreover, in page 21, as for “Quin et al. reported selenium-based far-red/ near-infrared luminogen…”, selenium-containing fluorescent probes was introduced again. Please carefully consider if these two sections can be put together.
- In page 21 ‘These fluorophores TPE-NIR, TPE-Blue, TPE-Crimson, TPE-Orange, and TPE-Red have been further explored in biological cell imaging Figure 19’ the author emphasized ‘biological cell imaging’ but this is not shown in Figure 19.
- The focus of the article is the design and application of the AIEgens, but the figures in the article focus too much on the molecular structure.
Author Response
This manuscript introduces AIEgens' latest developments in sensing, cell imaging, and Drug Delivery System. Depending on the relationship between the structure and function of AIEgens, the design strategies of AIEgens with different functions are summarized. This review is of significance for possible readers to understand the AIE materials in the field of biological system. However, this manuscript was written badly, and therefore I suggest a major revision before considering a publication of this review. Some concerns are given as follows.
- ‘Sensing’ has been applied in many fields, such as biological system in this review, environment, life, etc. This review focuses the sensing in biological system, and therefore, the authors should use “biological sensing” or “biosensing” in the title and subtitle.
Response: Thank you, hope the new title make a broad view
- In fact, there are many review papers summarizing AIE system, but what is the difference between this review and other reviews? The authors should introduce the difference in Introduction.
Response: Thank you, we have discussed at the end of Introductions. Hope you Ok with it, or otherwise suggest us to improve it.
- In page 2, I can understand that the abbreviation ‘RIV’ means ‘restriction of intramolecular vibration’ as marked in the picture, but the abbreviation is not mentioned in the text. Please check it.
- In page 5, does ‘The abnormal level of Ca in the body’ refer to the total calcium in the body or the Ca2+level in the body, please verify.
- For the reference [40] in the review (from page 5 to page 6), the manuscript introduced the synthesis method of SA-4CO2Na, but the mechanism for SA-4CO2Na detecting Ca2+ was missing. However, the corresponding Figure 5 shows the detection mechanism, so I advise to add the detection mechanism in the text.
- The source of many harmful ions and their detection methods are mentioned in page 8, but no cited reference is mentioned. Please add the corresponding reference to increase the credibility.
- In page 9, copper is an essential trace element for the human body and should not be simply classified as a toxic ion. Therefore, please confirm whether it is reasonable to classify copper ion detection as ‘toxic ion detection’.
- In pages 11 to 12, ‘Using the similar criteria of FR/NIR Qin and the group synthesized selenium-containing FR/NIR AIE active luminogen (TTSe dots) which are very rare however there are no reports available on selenium-containing fluorescent probes for bioimaging application.’ this sentence is vague and easily misunderstood. Moreover, in page 21, as for “Quin et al. reported selenium-based far-red/ near-infrared luminogen…”, selenium-containing fluorescent probes was introduced again. Please carefully consider if these two sections can be put together.
- In page 21 ‘These fluorophores TPE-NIR, TPE-Blue, TPE-Crimson, TPE-Orange, and TPE-Red have been further explored in biological cell imaging Figure 19’ the author emphasized ‘biological cell imaging’ but this is not shown in Figure 19.
- The focus of the article is the design and application of the AIEgens, but the figures in the article focus too much on the molecular structure.
Response: Thank you, for suggestions from point 3-10, we have done all the corrections and highlighted in the manuscript in yellow
Reviewer 3 Report
The manuscript “Recent Advances in Aggregation Induced Emission Active Materials for Sensing, Biological Cell Imaging and Drug Delivery System" by Bhosale et al. presents the recent advances in AIE active materials and their application in sensing, biological cell imaging, and drug delivery system. The review was constructed focused on different AIE-based materials, such as tetraphenylethylene, polymers, quantum dots, metal-organic frameworks, and so on. In my opinion, I believe that its content is well worth publishing in the Molecules. The review is well-written and well-organized. However, I would like to address a few questions to the authors. Based on my review I indicate minor revision. In this way, I have the following comments regarding the manuscript:
- I suggest the authors take care of oversized images (Figure 11, Figure 23, Figure 26, Figure 30) and chemical structures (Figure 10, Figure 11, Figure 18, Figure 22, Figure 28).
- Figure 4 presents so small carboxylates that the "-" seems a radical
- Figure 6 please take care about bonds and angles (PEN-TPE/PPL). In addition, I believe that "The structural illustration of" can be removed from the caption.
- Figure 15, I could not find Europium (III) in the presented compound.
- Figure 22, the "n" is confusing and could be differently presented
- TPE-MEM structure on page 18 must be presented as a Figure and cited in the text.
- I was wondering, did the authors evaluate to present AIE molecules for technological applications, such as OLEDs? (not mandatory)
- "mSiO2" by "mSiO2" (Page 24)
- Revise carefully the text for typos
Author Response
The manuscript “Recent Advances in Aggregation Induced Emission Active Materials for Sensing, Biological Cell Imaging and Drug Delivery System" by Bhosale et al. presents the recent advances in AIE active materials and their application in sensing, biological cell imaging, and drug delivery system. The review was constructed focused on different AIE-based materials, such as tetraphenylethylene, polymers, quantum dots, metal-organic frameworks, and so on. In my opinion, I believe that its content is well worth publishing in the Molecules. The review is well-written and well-organized. However, I would like to address a few questions to the authors. Based on my review I indicate minor revision. In this way, I have the following comments regarding the manuscript:
- I suggest the authors take care of oversized images (Figure 11, Figure 23, Figure 26, Figure 30) and chemical structures (Figure 10, Figure 11, Figure 18, Figure 22, Figure 28).
Response: Thank you, oversized images are corrected
- Figure 4 presents so small carboxylates that the "-" seems a radical
Response: Correction done. Thank you, done
- Figure 6 please take care about bonds and angles (PEN-TPE/PPL). In addition, I believe that "The structural illustration of" can be removed from the caption
Response: thank you, corrected
- Figure 15, I could not find Europium (III) in the presented compound.
Response: thank you, corrected
-Figure 22, the "n" is confusing and could be differently presented
Response: thank you, corrected
-TPE-MEM structure on page 18 must be presented as a Figure and cited in the text.
Response: thank you, corrected
- I was wondering, did the authors evaluate to present AIE molecules for technological applications, such as OLEDs? (not mandatory)
Response: We have not evaluated OLED application but yes, these AIE active molecules can be employed for OLED applications also.
- "mSiO2" by "mSiO2" (Page 24)
Response: thank you, corrected
- Revise carefully the text for typos
Response: Typos corrected
Round 2
Reviewer 1 Report
The authors have addressed the comments very well, more discussions about the decay channels of organic fluorescent materials and the rational design of AIEgens have been added. The typos have been corrected, and the manuscript has been improved significantly. Thus, the reviewer recommends the acceptance of the manuscript.
Reviewer 2 Report
This review can be accepted after minor revisions. The English should be polished. In addition, at the end of “Introduction”, I think that “…we decided to combining all the applications in one review article…” was a little exaggerated because AIE molecules have been studied in the field of optoelectronics (such as OLEDs) but were not involved in this review. This paragraph should be revised properly.